# Prefabrication of a ribosomal protein subcomplex essential for eukaryotic ribosome formation

Cohue Peña[1,2], Sabina Schütz[2], Ute Fischer[1], Yiming Chang[1], Vikram G Panse[2]*

[1]Institute of Biochemistry, ETH Zurich, Zurich, Switzerland; [2]Institute of Medical Microbiology, University of Zurich, Zurich, Switzerland

**Abstract** Spatial clustering of ribosomal proteins (r-proteins) through tertiary interactions is a striking structural feature of the eukaryotic ribosome. However, the functional importance of these intricate inter-connections, and how they are established is currently unclear. Here, we reveal that a conserved ATPase, Fap7, organizes interactions between neighboring r-proteins uS11 and eS26 prior to their delivery to the earliest ribosome precursor, the 90S. In vitro, uS11 only when bound to Fap7 becomes competent to recruit eS26 through tertiary contacts found between these r-proteins on the mature ribosome. Subsequently, Fap7 ATPase activity unloads the uS11:eS26 subcomplex onto its rRNA binding site, and therefore ensures stoichiometric integration of these r-proteins into the 90S. Fap7-depletion in vivo renders uS11 susceptible to proteolysis, and precludes eS26 incorporation into the 90S. Thus, prefabrication of a native-like r-protein subcomplex drives efficient and accurate construction of the eukaryotic ribosome.

## Introduction

Ribosomes are the universal molecular machines responsible for the final step of translating genetic information into proteins. In eukaryotes, the large 60S subunit contains three rRNAs (25S, 5.8S, 5S) and 46 r-proteins. The small 40S subunit contains a single rRNA (18S) and 33 r-proteins (*Klinge et al., 2011*; *Rabl et al., 2011*). A coordinated effort of all three transcriptional machineries (RNA polymerases I, II and III) and >200 conserved assembly factors drives eukaryotic ribosome production (*de la Cruz et al., 2015*; *Woolford and Baserga, 2013*).

RNA polymerase I driven production of 35S pre-rRNA initiates eukaryotic ribosome assembly in the nucleolus. The emerging pre-rRNA associates with small subunit r-proteins, and ~100 assembly factors to form the earliest precursor, the 90S (*Dragon et al., 2002*; *Grandi et al., 2002*; *Kos and Tollervey, 2010*). Cleavage within the 35S pre-rRNA releases the small subunit precursor, the 40S pre-ribosome and permits the remaining pre-rRNA to associate with 60S-r-proteins and assembly factors to form the 60S pre-ribosome. A 40S pre-ribosome undergoes few compositional changes as it travels through the nucleoplasm and is rapidly exported into the cytoplasm (*Grandi et al., 2002*; *Schäfer et al., 2003*). In contrast, a 60S pre-ribosome transiently interacts with ~80 assembly factors, as it travels towards the nuclear periphery (*Bassler et al., 2001*; *Fatica et al., 2002*; *Harnpicharnchai et al., 2001*; *Nissan et al., 2002*). At distinct stages, assembly factors are released from ribosome precursors possibly through the action of >50 energy-consuming RNA helicases, AAA-ATPases, ABC-ATPases and GTPases (*Kressler et al., 2010*; *Panse and Johnson, 2010*; *Strunk and Karbstein, 2009*). The precise targets for many of these diverse energy-consuming enzymes remain to be elucidated.

Pre-ribosomal particles are transported by multiple transport receptors through nuclear pore complexes (NPCs) into the cytoplasm where they undergo final maturation before initiating

*For correspondence: vpanse@ imm.uzh.ch

**Competing interests:** The authors declare that no competing interests exist.

translation (*Sloan et al., 2016*). Cytoplasmic maturation of the 40S pre-ribosome requires endonucleolytic cleavage of the 20S pre-rRNA at site D to generate the 3' end of mature 18S rRNA. This step is catalyzed by the endonuclease Nob1 (*Fatica et al., 2004*; *Lamanna and Karbstein, 2009*; *Pertschy et al., 2009*) and is thought to occur within an 80S-like particle formed upon the interaction of a 40S pre-ribosome with a mature 60S subunit (*Lebaron et al., 2012*; *Strunk et al., 2012*). Functional studies have identified additional factors that influence the processing of immature 20S pre-rRNA. These include the Nob1-interacting protein Pno1, the methyltransferase Dim1, the GTPase mimic Tsr1, atypical ATPases Rio1 and Rio2, the DEAH ATPase Prp43 and its cofactor Pfa1 (*Ferreira-Cerca et al., 2007, 2014*; *Geerlings et al., 2003*; *McCaughan et al., 2016*; *Pertschy et al., 2009*; *Strunk et al., 2011*; *Turowski et al., 2014*). Additionally, r-proteins uS11 and eS26 (yeast Rps14a/b and Rps26a/b; nomenclature according to *Ban et al., 2014*) are essential for endonucleolytic cleavage of 20S pre-rRNA (*Jakovljevic et al., 2004*; *Schütz et al., 2014*). eS26 clamps the 3' end of 18S rRNA and directly contacts its neighboring r-protein uS11. Depletion of Tsr2 or the ATPase Fap7, the binding partners of eS26 and uS11, respectively, also impairs cytoplasmic processing of 20S pre-rRNA (*Granneman et al., 2005*; *Schütz et al., 2014*; *Strunk et al., 2012*), suggesting that these r-proteins play a role to correctly position the D site for Nob1-mediated 20S pre-rRNA cleavage.

Eukaryotic ribosome assembly relies on rapid nucleocytoplasmic transport. In yeast, within 90 min, the majority of ~14 million newly synthesized r-proteins are targeted to the nuclear compartment by the import machinery for incorporation into pre-ribosomal particles (*Warner, 1999*). r-proteins contain large unstructured regions that are prone to non-specific interactions with nucleic acids and proteolytic degradation in their non-assembled state (*Jäkel and Görlich, 1998*; *Jäkel et al., 2002*). To overcome this logistical challenge, a subset of newly synthesized r-proteins employ dedicated chaperones that bind them in the cytoplasm and co-coordinate their nuclear import (*Mitterer et al., 2016*; *Pausch et al., 2015*; *Pillet et al., 2015*; *Stelter et al., 2015*). Recently, we reported a different mechanism by which the r-protein eS26 is safely transferred to the 90S (*Schütz et al., 2014*). In this case, eS26, like a typical import cargo, directly recruits multiple importins for its transport to the nucleus. However, unlike a typical import cargo, after reaching the nuclear compartment, eS26 is extracted from the importin by an unloading factor, the escortin Tsr2, without the aid of RanGTP. Once bound to Tsr2, eS26 is shielded from proteolysis, enabling its safe transfer to the 90S.

Another challenge for the assembly machinery is to ensure that those r-proteins, imported into the nucleus, are correctly integrated into each pre-ribosome, before the irreversible export step. How the assembly machinery co-ordinates incorporation of different r-proteins into their rRNA binding site(s), and thereby prevents wasteful production of incompletely assembled ribosomes is currently unclear. In this study, we unveil an essential energy-dependent mechanism that enables stoichiometric integration of r-proteins into an assembling 90S. We demonstrate that Fap7, a conserved ATPase, pre-organizes interactions between neighboring r-proteins uS11 and eS26 through native-like tertiary contacts prior to their delivery to the 90S. Subsequently, Fap7 ATPase activity concurrently integrates the uS11:eS26 subcomplex into the 90S. Thus, prefabrication of an uS11:eS26 subcomplex guarantees stoichiometric levels of these r-proteins are incorporated into the 90S.

## Results

### Co-overexpression of Fap7 and the r-protein uS11 bypasses in vivo requirement for Tsr2

Tsr2 is an evolutionarily conserved 23.7 kDa escortin that unloads the r-protein eS26 from its import receptor after entering the nucleus in a RanGTP-independent manner. After unloading, Tsr2 binds the released eS26 and ensures its safe transfer to the earliest ribosome precursor, the 90S (*Schütz et al., 2014*). While eS26 is essential for yeast cell viability, Tsr2-depleted cells are viable, but severely impaired in growth (*Figure 1A*; *Schütz et al., 2014*). These data implicate a Tsr2-independent mechanism that safely delivers the r-protein to the 90S to sustain viability of a Tsr2-depleted strain.

To reveal this alternative pathway, we employed a genetic approach. A high-copy suppressor screen was performed in the model organism yeast, to identify genes, which, upon overexpression,

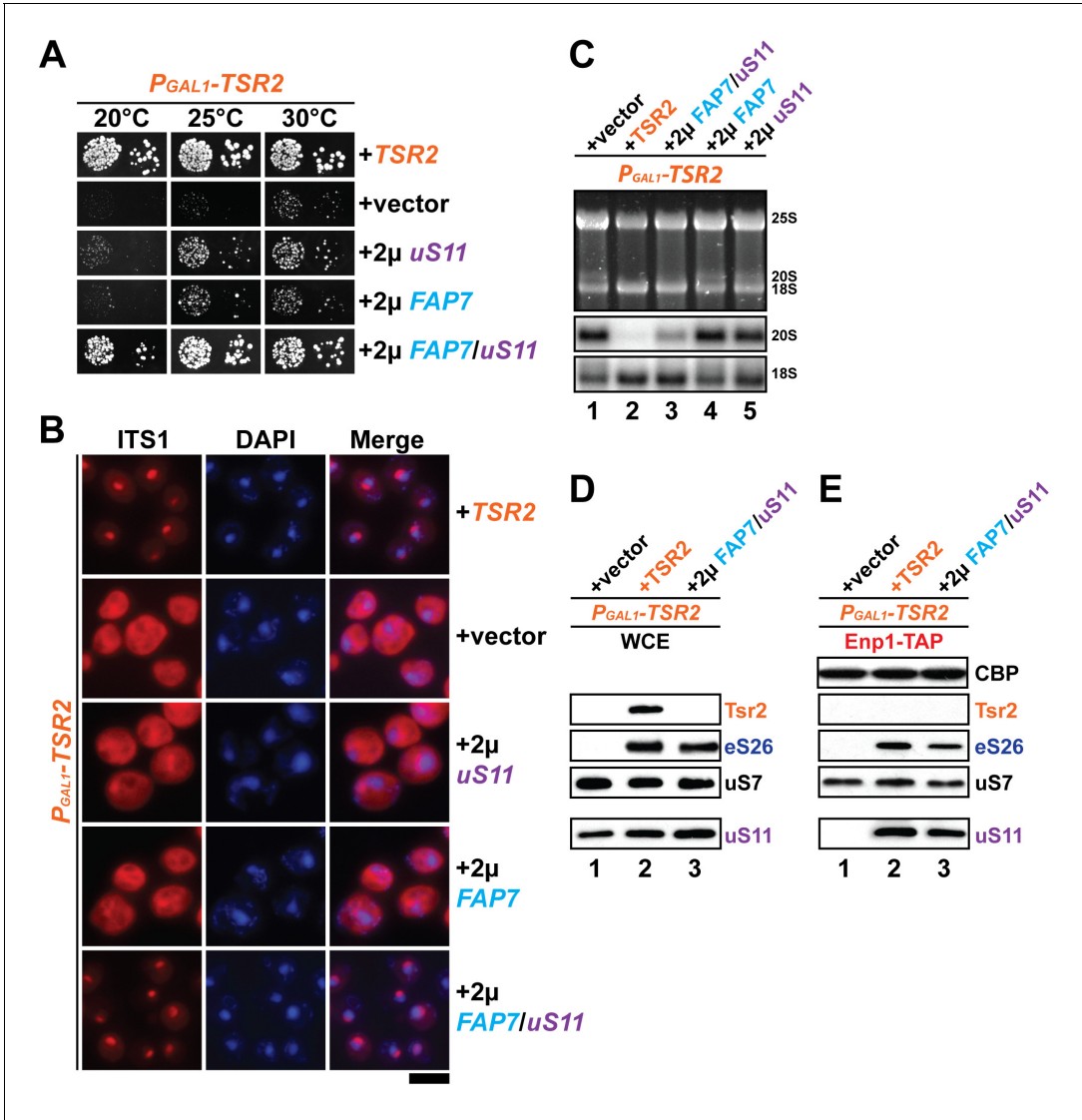

**Figure 1.** Co-overexpression of Fap7 and the r-protein uS11 bypasses in vivo requirement for Tsr2. (**A**) Co-overexpression of *FAP7* and *uS11* rescues slow growth of Tsr2-depleted cells. The P<sub>GAL1</sub>-*TSR2* strain was transformed with indicated plasmids and spotted in 10-fold dilutions on selective and repressive glucose-containing plates and grown at indicated temperatures for 3–7 days. (**B**) Co-overexpression of *FAP7* and *uS11* rescues 20S pre-rRNA processing defect of Tsr2-depleted cells. Indicated strains were grown to mid-log phase at 25°C in selective glucose-containing medium. 20S pre-rRNA was localized by FISH using a Cy3-labeled oligonucleotide complementary to the 5' portion of ITS1 (red). Nuclear and and mitochondrial DNA was stained with DAPI (blue). Scale bar = 5 µm. (**C**) Indicated strains were grown as in above for extraction of total RNA and then analyzed by GelRed straining and Northern blotting using probes against 20S and 18S rRNAs. (**D**) Co-overexpression of *FAP7* and *uS11* restores protein levels of eS26 in Tsr2-depleted cells. Whole cell extracts (WCE) were prepared from indicated strains and subjected to Western analysis. (**E**) Co-overexpression of *FAP7* and *uS11* restores co-enrichment of eS26 with Enp1-TAP in Tsr2-depleted cells. Enp1-TAP was isolated from indicated strains and subjected to Western analysis. uS7 and CBP (TAP-tag) protein levels served as loading controls.

suppressed the severe growth defect of a *TSR2* conditional mutant (P<sub>GAL1</sub>-*TSR2*). To this end, the P<sub>GAL1</sub>-*TSR2* mutant strain was transformed with a high-copy (2µ) plasmid library and grown on repressive glucose-containing media at 25°C. Plasmids were recovered from fast-growing suppressor colonies. Sequence analyses identified *RPS14a* (hereafter termed *uS11*), a gene encoding the 40S subunit r-protein uS11 (yeast Rps14), and another gene *YDL166C* encoding the essential ATPase Fap7 as candidate suppressors. To directly test this possibility, *FAP7* and *uS11* were subcloned into high-copy plasmids and then tested for their ability to suppress the impaired growth of the P<sub>GAL1</sub>-*TSR2* mutant. These analyses revealed that overexpression of *FAP7* and *uS11* weakly rescued the

growth defect of the P$_{GAL1}$-*TSR2* mutant, as determined by the size of single colonies (*Figure 1A*). Interestingly, uS11 forms a stable complex with the ATPase Fap7 (*Granneman et al., 2005*). Structural studies of the Fap7:uS11 complex revealed that Fap7 functions as a RNA mimic suggesting that it stabilizes uS11 before being incorporated into 40S ribosomes (*Hellmich et al., 2013*; *Loc'h et al., 2014*). Co-overexpression of *FAP7* and *uS11* together restored impaired growth of the P$_{GAL1}$-*TSR2* mutant to nearly WT rates (*Figure 1A*).

Next, we investigated whether co-overexpression of *FAP7* and *uS11* rescued different phenotypes exhibited by Tsr2-depleted cells (*Peng et al., 2003*; *Schütz et al., 2014*). First, we assessed whether impaired cytoplasmic processing of 20S pre-rRNA seen in the P$_{GAL1}$-*TSR2* mutant was rescued upon *FAP7* and *uS11* co-overexpression. For this, we localized the 5′ portion of the internal transcribed spacer 1 (ITS1) by fluorescence in situ hybridization (FISH). ITS1 is present within immature 20S pre-rRNA, but not in mature 18S rRNA. In a WT strain, due to efficient nuclear export of 40S pre-ribosomes, Cy3-ITS1 (red) is detectable only in the nucleolus (*Figure 1B*). After nuclear export, ITS1 is rapidly cleaved from 20S pre-rRNA by the endonuclease Nob1 and degraded by the exonuclease Xrn1 (*Moy and Silver, 1999*; *Stevens et al., 1991*). As previously reported, Tsr2-depleted cells accumulate Cy3-ITS1 in the cytoplasm (*Figure 1B*; *Schütz et al., 2014*), indicating impairment in cytoplasmic 20S pre-rRNA processing. However, these cells co-overexpressing *FAP7* and *uS11* did not show this phenotype as judged by the absence of cytoplasmic fluorescence signal of Cy3-ITS1 (*Figure 1B*). Next, we assessed 20S pre-rRNA levels by Northern blotting using an oligonucleotide complementary to ITS1. Consistent with previous studies, P$_{GAL1}$-*TSR2* mutant cells showed high levels of immature 20S pre-rRNA (*Figure 1C* lane 1; *Peng et al., 2003*; *Schütz et al., 2014*). Co-overexpressing *FAP7* and *uS11* significantly reduced 20S pre-RNA levels in these cells (*Figure 1C*, compare lanes 1 and 3). These data indicate that co-overexpression of *FAP7* and *uS11* rescues cytoplasmic 20S pre-rRNA processing defects observed in the Tsr2-depleted cells.

Another phenotype associated with the P$_{GAL1}$-*TSR2* mutant is the strongly reduced eS26 protein levels in vivo (*Figure 1D*). Co-overexpression of *FAP7* and *uS11* in Tsr2-depleted cells restored eS26 protein levels to nearly WT levels as judged by Western analyses of whole cell extracts (WCEs) (*Figure 1D*, compare lanes 1 and 3).

Finally, we evaluated eS26 co-enrichment with early pre-ribosomes in the P$_{GAL1}$-*TSR2* mutant cells co-overexpressing *FAP7* and *uS11*. For this, we isolated Enp1-TAP that purifies both the 90S and an early 40S pre-ribosome from these cells (*Grandi et al., 2002*; *Schäfer et al., 2003*). As previously observed, eS26 co-enrichment with Enp1-TAP isolated from Tsr2-depleted cells was strongly impaired (*Figure 1E*, lane 1; *Schütz et al., 2014*). Co-overexpression of *FAP7* and *uS11* in these cells restored eS26 co-enrichment with Enp1-TAP to nearly WT levels (*Figure 1E*, compare lanes 1 and 3).

Based on all these studies, we conclude that co-overexpression of *FAP7* and *uS11* compensates Tsr2-requirement in safely delivering eS26 to the 90S.

## Incorporation of uS11 and eS26 into pre-ribosomes is interdependent

The nuclear localized Fap7 (*Figure 2A*) was initially isolated in a genetic screen as a factor required for Pos9-driven transcription of genes upon oxidative stress (*Juhnke et al., 2000*). Subsequently, it was identified as an energy-consuming enzyme that is not a structural component of pre-ribosomes, but directly binds the r-protein uS11 (*Granneman et al., 2005*). Consistent with these studies, we found that uS11 alone, but not Fap7, co-enriched with the 90S and 40S pre-ribosomes (*Figure 2B*). Moreover, like the escortin Tsr2, Fap7-depletion did not impair export of 40S pre-ribosomes (*Figure 2C*; *Schütz et al., 2014*). However, these exported pre-ribosomes failed to process immature 20S pre-rRNA into mature 18S rRNA (*Figure 2D*; *Granneman et al., 2005*).

While co-overexpression of *FAP7* and *uS11* restored impaired growth of Tsr2-depleted cells to nearly WT rates, co-overexpression of *TSR2* and *RPS26a*, did not rescue the lethality of the Fap7-depletion strain (*Figure 2E*). These genetic data led us to wonder whether the Fap7/uS11 pair functions downstream of Tsr2 to transfer eS26 to the 90S. We evaluated eS26 and uS11 co-enrichment with Enp1-TAP isolated from Fap7-depleted cells (*Figure 2F*) and Tsr2-depleted cells (*Figure 2G*). Western analyses revealed that both uS11 and eS26 poorly co-enriched with the Enp1-TAP isolated from Fap7-depleted cells (*Figure 2F*) and from Tsr2-depleted cells (*Figure 2G*). In contrast, r-proteins uS3 and uS7 efficiently co-enriched with Enp1-TAP isolated from Fap7-depleted (*Figure 2F*) and Tsr2-depleted cells (*Figure 2G*). We conclude that incorporation of eS26 and uS11 into the pre-ribosome requires both Fap7 and Tsr2.

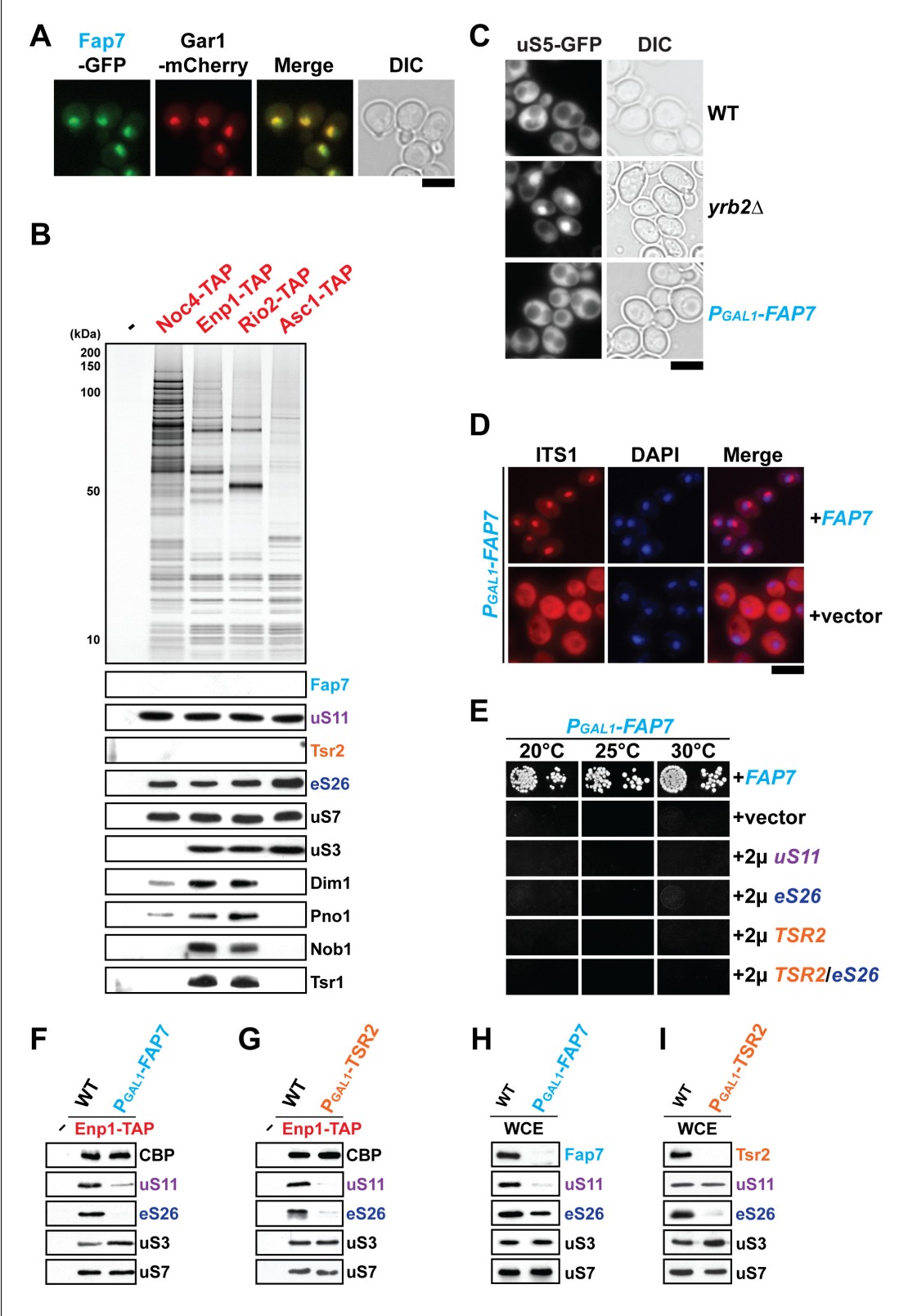

**Figure 2.** uS11 and eS26 depend upon each other for their incorporation into pre-ribosomes. (**A**) Fap7 is a nuclear localized protein. Fap7 and the nucleolar protein Gar1 were endogenously tagged with GFP and mCherry, respectively, and then strains expressing them were grown to mid-log phase at 25°C. Localization of Fap7-GFP and Gar1-mCherry was visualized by fluorescence microscopy. Scale bar = 5 μm. (**B**) uS11 and eS26, but not Fap7 or Tsr2 co-enrich with pre-ribosomal particles along the 40S maturation pathway. Pre-ribosomal particles in the 40S maturation pathway were purified

*Figure 2 continued on next page*

*Figure 2 continued*

using the indicated TAP-tagged baits. Calmodulin-eluates were analyzed by Silver staining, and Western analyses was performed using the indicated antibodies. The r-protein uS7 served as loading control for the TAPs. (C) Fap7-depletion does not impair pre-40S subunit nuclear export. The indicated strains expressing uS5-GFP were grown in repressive glucose-containing medium to mid-log phase at 25°C. Localization of uS5-GFP was monitored by fluorescence microscopy. Scale bar = 5 μm. (D) Fap7-depleted cells accumulate immature 20S pre-rRNA in the cytoplasm. Indicated strains were grown to mid-log phase at 25°C in selective glucose-containing medium. 20S pre-rRNA was localized by FISH using a Cy3-labeled oligonucleotide complementary to the 5' portion of ITS1 (red). Nuclear and mitochondrial DNA was stained with DAPI (blue). Scale bar = 5 μm. (E) Slow growth of Fap7-depleted cells cannot be rescued by either (co-)overexpression of *uS11*, *eS26* or *TSR2*. The P$_{GAL1}$-*FAP7* strain was transformed with indicated plasmids and spotted in 10-fold dilutions on selective and repressive glucose-containing plates and grown at indicated temperatures for 3–7 days. (F) Efficient recruitment of uS11 and eS26 to the 90S requires Fap7. Enp1-TAP was isolated from indicated strains and subjected to Western analysis. (G) Efficient recruitment of uS11 and eS26 to the 90S requires Tsr2. Enp1-TAP was isolated from indicated strains and subjected to Western analysis. Protein levels of uS7 and CBP (TAP-tag) served as loading control. (H) uS11, but not eS26 levels are strongly reduced in Fap7-depleted cells. (I) eS26, but not uS11 levels are strongly reduced in Tsr2-depleted cells. Whole cell extracts (WCE) were prepared from indicated strains and subjected to Western analysis. uS7 protein levels served as a loading control.

Next, we assessed uS11 and eS26 protein levels in WCEs derived from Fap7-depleted and from Tsr2-depleted cells by Western blotting. Fap7-depletion strongly reduced uS11 protein levels in WCEs, but eS26 protein levels were very similar to WT levels (*Figure 2H*). These data show that Fap7-depletion does not affect eS26 protein stability in vivo, but impairs its targeting to the pre-ribosome (*Figure 2F and H*). Consistent with previous studies, Tsr2-depleted cells showed strongly reduced levels of eS26 in WCEs (*Schütz et al., 2014*). uS11 protein levels were very similar to WT levels in these extracts (*Figure 2I*). These data show that Tsr2-depletion does not influence uS11 protein stability in vivo, but impairs its targeting to the pre-ribosome (*Figure 2G and I*). Notably, co-overexpression of *FAP7* and *uS11* in Tsr2-depleted cells that restored eS26 protein levels and co-enrichment with pre-ribosomes (*Figure 1D and E*) also restored uS11 co-enrichment with Enp1-TAP to nearly WT levels (*Figure 1E*, compare lanes 1 and 3).

Based on all these studies, we suggest that uS11 and eS26 are dependent upon each other for their incorporation into the 90S.

## The Fap7:uS11 complex efficiently recruits eS26

The requirement of both Fap7 and Tsr2 to target uS11 and eS26 to the pre-ribosome led us to test physical interactions between Fap7, uS11, Tsr2 and eS26. To this end, we performed in vitro binding assays using purified recombinant proteins (*Figure 3A*). Immobilized GST-Fap7 and GST-Fap7:uS11 complex were incubated with recombinant eS26 and Tsr2:eS26 complex in the presence of competing *Escherichia coli* lysates (*Figure 3A*). These studies revealed that a preformed GST-Fap7:uS11 complex efficiently recruited eS26 (*Figure 3A*, lane 2). GST-Fap7 alone did not recruit eS26 (*Figure 3A*, lane 5). Neither GST-Fap7 nor GST-Fap7:uS11 interacted with the Tsr2:eS26 complex (*Figure 3A*, lane 3 and 6). GST-uS11 did not interact with eS26 and the Tsr2:eS26 complex (*Figure 3A*, lane 11 and 12). Based on these studies, we conclude that uS11 only in complex with Fap7 is competent to recruit eS26.

Next, we characterized the interaction between the GST-Fap7:uS11 complex and eS26. On the mature 40S subunit, uS11 and eS26 are neighbors and contact each other through a network of conserved salt-bridges and hydrogen bonds (*Figure 3B*). These include: (1) R103 / R107 on uS11 and D52 on eS26, (2) R114 on uS11 and Y59 / Y62 on eS26. We tested whether these contacts contribute to recruiting eS26 to the Fap7:uS11 complex in vitro. To this end, we constructed an uS11 triple mutant wherein three arginine residues (R103, R107 and R114) were mutated to aspartates, hereafter termed uS11$^{3R}$. The uS11$^{3R}$ mutant protein was not impaired in binding Fap7 (*Figure 3C*, lane 2). However, the mutant GST-Fap7:uS11$^{3R}$ heterodimer was significantly impaired in recruiting eS26 (*Figure 3C*, lane 2). These data support the notion that eS26 recruitment to the Fap7:uS11 complex involves native-like tertiary contacts found between these r-proteins on the mature 40S subunit.

We wondered whether these conserved tertiary contacts are critical to suppress the growth defect of the P$_{GAL1}$-*TSR2* mutant cells. Therefore, we co-overexpressed *FAP7* and *uS11*$^{3R}$ in Tsr2-depleted cells and analyzed growth of the resulting strain (*Figure 3D*). Co-overexpression of *FAP7* and *uS11*$^{3R}$ did not rescue the growth and different P$_{GAL1}$-*TSR2*-associated phenotypes such as

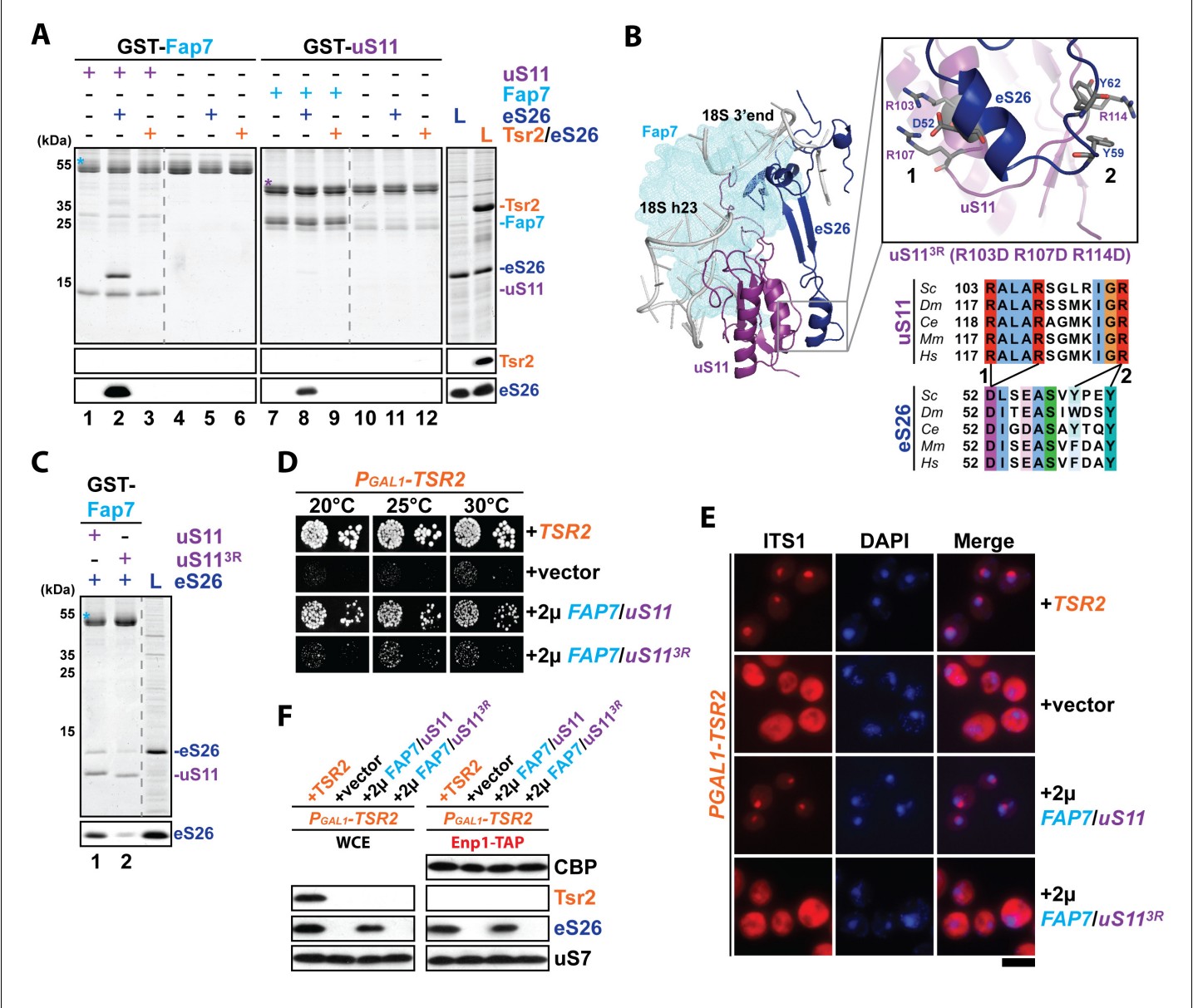

**Figure 3.** The Fap7:uS11 complex recruits eS26. (**A**) The Fap7:uS11 complex directly binds eS26 but not Tsr2:eS26 in vitro. Recombinant GST-tagged Fap7 and uS11 complexes were immobilized on Glutathione Sepharose beads before incubation with *E. coli* lysate containing recombinant eS26 or Tsr2:eS26. After washing away unbound proteins, beads were eluted and analyzed by SDS-PAGE followed by Coomassie Blue staining and Western blotting. L = 10% input. GST-baits are indicated with asterisks. (**B**) Conserved protein-protein interactions between uS11 and eS26 on the mature 40S subunit. uS11 (purple) binds to 18S rRNA helix 23 (grey) with its C-terminus embedded into the 18S rRNA 3'end. eS26 (blue) forms major contacts with uS11 via (1) a salt-bridge between uS11 R103 / R107 and eS26 D52 and (2) a hydrophobic side-chain interaction between uS11 R114 and eS26 Y59 / Y62. Yeast Fap7 which acts as a RNA mimic for helix 23 (light blue) was modeled into the structure by superposition of the Fap7:uS11 complex from *Pyrococcus abyssi* (PDB: 4CVN; *Loc'h et al., 2014*) onto yeast uS11 from the mature 40S ribosome (PDB: 4 V88; *Ben-Shem et al., 2011*). For the uS11^3R mutant, R103 / R107 / R114 were each mutated to aspartates. The sequences for the following organisms were aligned: *Saccharomyces cerevisiae (Sc)*, *Drosophila melanogaster (Dm)*, *Caenorhabditis elegans (Ce)*, *Mus musculus (Mm)* and *Homo sapiens (Hm)* (**C**) The uS11^3R mutant is impaired in recruiting eS26 in vitro. Recombinant GST-Fap7:uS11 or GST-Fap7:uS11^3R was incubated with eS26 and analyzed by SDS-PAGE followed by Coomassie Blue staining and Western blotting. L = 100% input. GST-baits are indicated with asterisks. (**D**) Co-overexpression of *FAP7* and *uS11^3R* does not rescue the slow growth of Tsr2-depleted cells. The P_GAL1-*TSR2* strain was transformed with indicated plasmids and spotted in 10-fold dilutions on selective and repressive glucose-containing plates and grown at indicated temperatures for 3–7 days. (**E**) Co-overexpression of *FAP7* and *uS11^3R* does not rescue the 20S pre-rRNA processing defect of Tsr2-depleted cells. Indicated strains were grown to mid-log phase at 25°C in selective glucose-containing medium. 20S pre-rRNA was localized by FISH using a Cy3-labeled oligonucleotide complementary to the 5' portion of ITS1 (red). Nuclear and and mitochondrial DNA was stained with DAPI (blue). Scale bar = 5 μm. (**F**) Co-overexpression of *FAP7* and *uS11^3R* does not restore protein levels of eS26 and neither

*Figure 3 continued on next page*

*Figure 3 continued*

allows co-enrichment with Enp1-TAP in Tsr2-depleted cells. Whole cell extracts (WCE, left panel) and Enp1-TAP (right panel) were prepared and isolated, respectively, from indicated strains and subjected to Western analysis. uS7 and CBP (TAP-tag) protein levels served as loading controls.

The following figure supplement is available for figure 3:

**Figure supplement 1.** r-proteins cluster on the eukaryotic ribosome via tertiary contacts.

impaired 20S pre-rRNA processing, eS26 protein levels and recruitment to the pre-ribosome (*Figure 3E and F* and *Figure 4—figure supplement 1*).

Based on these functional studies, we conclude that robust interactions between uS11 and eS26 are critical to stabilize and deliver eS26 to the 90S, when Tsr2 is absent.

## Fap7 ATPase activity is required to bypass Tsr2 requirement

We investigated whether Fap7 ATPase activity is critical to suppress the growth defect of the P$_{GAL1}$-*TSR2* mutant. To test this, we constructed a *fap7* mutant, hereafter termed *fap7-2*, in which the two conserved aspartate and histidine residues D82 and H84 within the Walker B motif were mutated to alanines (*Figure 4A*, left panel; *Granneman et al., 2005*; *Strunk et al., 2012*). Next, we characterized the ATPase activity of Fap7-2 mutant protein. For this, we monitored stimulation of ATPase activity of Fap7 and the Fap7-2 mutant protein using a coupled-enzyme assay that detects formation of ADP upon ATP hydrolysis (*Montpetit et al., 2012*). Consistent with previous studies (*Hellmich et al., 2013*; *Loc'h et al., 2014*), Fap7 alone shows no significant ATPase activity (*Figure 4B*). However, binding to uS11 stimulates its ATPase activity (*Hellmich et al., 2013*) as indicated by an increase in ADP formation over time. While, the Fap7-2 Walker B mutant bound uS11 efficiently (*Figure 4A*, right panel), its ATPase activity could not be stimulated and was very similar to Fap7 activity alone (*Figure 4B*).

Next, we investigated whether the Fap7-2 Walker B mutant is impaired in ATP binding using fluorescence-labeled mant-ATP (*Manikas et al., 2016*). These analyses revealed that the Fap7-2 Walker B mutant is severely impaired in ATP binding, as its fluorescence emission spectrum, unlike wild-type Fap7, remained similar to mant-ATP alone (*Figure 4C*). Thus, the impaired ATPase activity of the Fap7-2 mutant seems to arise from its inability to bind ATP.

We then tested whether the Fap7-2 mutant together with uS11 is able to compensate the functional requirement of Tsr2. Thus, we analyzed the growth of P$_{GAL1}$-*TSR2* mutant cells co-overexpressing *fap7-2* and *uS11*. These analyses revealed that co-overexpression of *fap7-2* and *uS11* was unable to rescue the growth impairment of the P$_{GAL1}$-*TSR2* mutant (*Figure 4D*). As expected, these cells were still impaired in cytoplasmic 20S pre-rRNA processing and recruitment of eS26 to pre-ribosomes (*Figure 4E and F* and *Figure 4—figure supplement 1*). Based on all these data, we conclude that co-overexpression of *uS11* and a catalytically active *FAP7* is critical to compensate the functional requirement of Tsr2 in vivo.

## Fap7 ATPase activity organizes and recruits the uS11:eS26 subcomplex to helix 23 of 18S rRNA

Two studies had revealed that the catalytic activity of Fap7 regulates the interaction of uS11 with its 18S rRNA binding site helix 23 (H23) in vitro (*Hellmich et al., 2013*; *Loc'h et al., 2014*). We tested whether Fap7 ATPase activity is required to load a preformed uS11:eS26 complex onto H23 rRNA. For this, immobilized GST-uS11 was incubated with eS26 and H23 rRNA in the presence of Fap7 or mutant Fap7-2, and ATP or the non-hydrolysable ATP-analog AMPPNP (summarized in *Figure 5A*). These interaction studies revealed that Fap7 and eS26 were efficiently recruited to GST-uS11 in the absence of ATP (*Figure 5B*, lane 4). Consistent with the reported RNA mimic function for Fap7 (*Loc'h et al., 2014*), H23 rRNA was hardly recruited to the Fap7:uS11:eS26 complex in the absence of ATP (*Figure 5B*, lane 8). Notably, GST-uS11 was efficient in loading eS26 and H23 rRNA in the presence of catalytically active Fap7 and ATP (*Figure 5B*, lane 9). GST-uS11 failed to recruit H23 rRNA in the presence of a catalytically inactive Fap7-2 mutant, and Fap7 loaded with the non-hydrolysable ATP analog AMPPNP (*Figure 5B*, lanes 10–13). Recruitment of H23 rRNA to the GST-uS11:

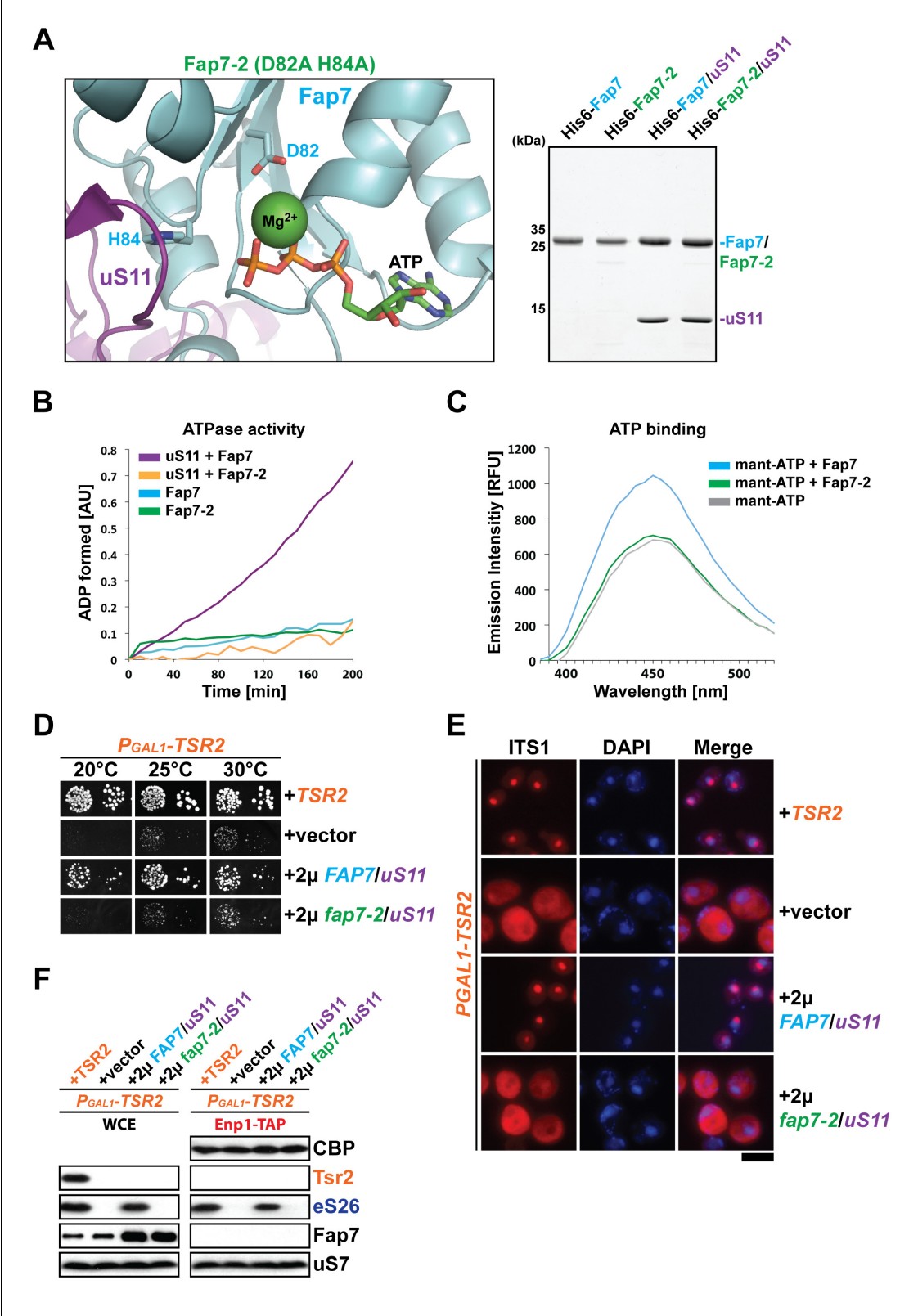

**Figure 4.** Fap7 ATPase activity is required to bypass Tsr2 requirement. (**A**) To generate a catalytically inactive ATPase mutant for in vitro assays, the conserved D82 and H84 residues in the Walker B motif of Fap7 were each mutated to alanine (left panel), and expressed and purified from *E. coli* (right panel). The structure shows the Fap7:uS11 complex from *P. abyssi* in complex with ATP and Mg²⁺ (PDB: 4CW7; *Loc'h et al., 2014*). (**B**) Fap7-2 is deficient in ATP hydrolysis. The ATPase activity of indicated Fap7 and uS11 complexes was monitored by an indirect enzyme assay that detects

*Figure 4 continued on next page*

*Figure 4 continued*

formation of ADP, which is then coupled to β-NADH oxidation via the action of pyruvate kinase (PK) and lactate dehydrogenase (LDH). The decrease in β-NADH absorbance over time was measured at 340 nm in arbitrary units (AU). (C) Fap7-2 is deficient in ATP binding. Fluorescent mant-ATP was mixed with purified Fap7 or Fap7-2 and the change in emission spectra upon nucleotide binding was measured in relative fluorescence units (RFUs). (D) Co-overexpression of *fap7-2* and *uS11* does not rescue the slow growth of Tsr2-depleted cells. The $P_{GAL1}$-*TSR2* strain was transformed with indicated plasmids and spotted in 10-fold dilutions on selective and repressive glucose-containing plates and grown at indicated temperatures for 3–7 days. (E) Co-overexpression of *fap7-2* and *uS11* does not rescue the 20S pre-rRNA processing defect of Tsr2-depleted cells. Indicated strains were grown to mid-log phase at 25°C in selective glucose-containing medium. Localization of 20S pre-rRNA was analyzed by FISH using a Cy3-labeled oligonucleotide complementary to the 5' portion of ITS1 (red). Nuclear and and mitochondrial DNA was stained with DAPI (blue). Scale bar = 5 μm. (F) Co-overexpression of *fap7-2* and *uS11* does not restore protein levels of eS26 and neither allows co-enrichment with Enp1-TAP in Tsr2-depleted cells. Whole cell extracts (WCE, left panel) and Enp1-TAP (right panel) were prepared and isolated, respectively, from indicated strains and subjected to Western analysis. uS7 and CBP (TAP-tag) protein levels served as loading controls.

The following figure supplement is available for figure 4:

**Figure supplement 1.** Co-overexpression of FAP7 and uS113R or fap7-2 and uS11 does not rescue the 20S pre-rRNA processing defect of Tsr2-depleted cells.

eS26 complex is not due to bound eS26, since GST-eS26 alone did not detectably bind H23 rRNA (*Figure 5C*, lane 2). Further, nearly identical levels of H23 rRNA were recruited to GST-uS11 and GST-uS11:eS26 complexes, in the presence of Fap7 and ATP (*Figure 5C*, compare lanes 5 and 7). Altogether, these data show that Fap7 orchestrates the formation of an uS11:eS26 complex, and its ATPase activity facilitates loading of this prefabricated r-protein subcomplex onto H23 rRNA.

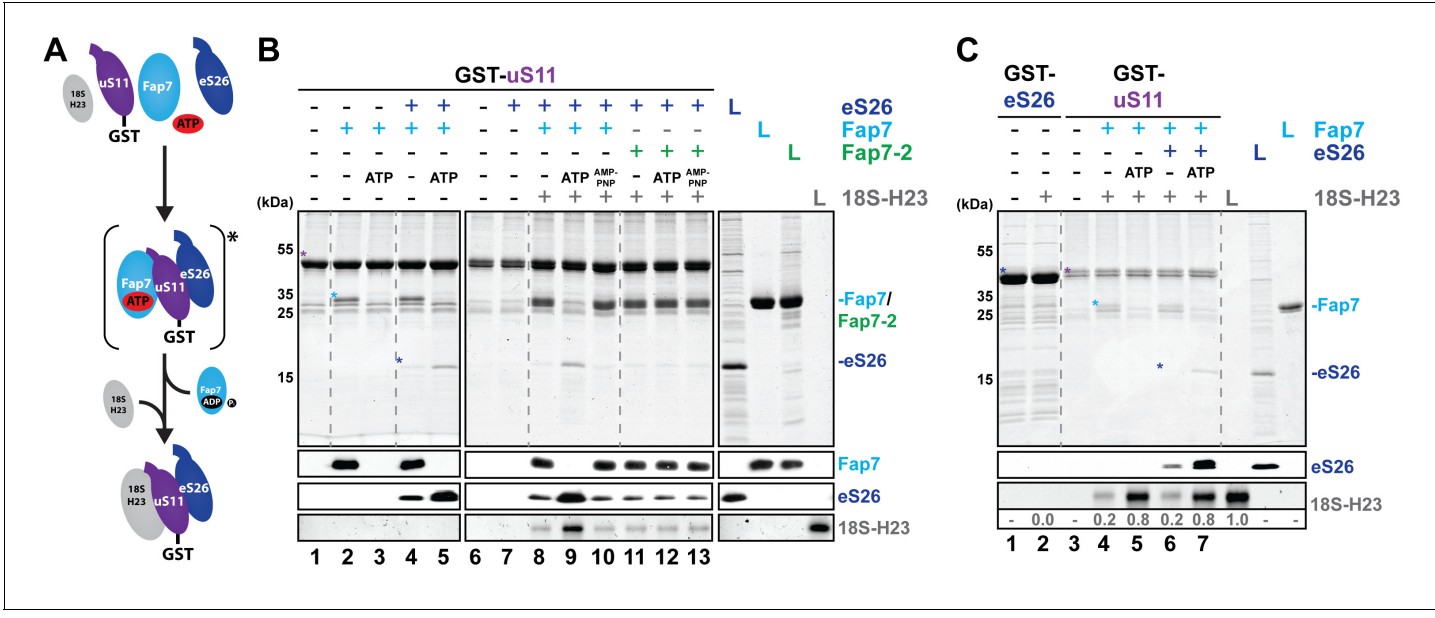

**Figure 5.** Fap7 ATPase activity organizes and recruits the uS11:eS26 subcomplex to helix 23 of 18S rRNA. (A) Schematic for ATP-dependent loading of uS11:eS26 onto helix 23 of 18S rRNA by Fap7. Asterisk indicates a potential intermediate Fap7-ATP:uS11:eS26 complex. (B) Recombinant GST-uS11 was immobilized on Glutathione Sepharose beads before incubation with Fap7 or Fap7-2, eS26, ATP or non-hydrolyzable ATP analog AMP-PNP and 18S helix 23 rRNA. (C) Recombinant GST-eS26 was immobilized on Glutathione Sepharose beads before incubation with 18S helix 23 rRNA. After washing away unbound proteins and RNA, beads were eluted and analyzed by SDS-PAGE followed by Coomassie Blue staining and Western blotting. For analysis of RNA, samples were Phenol-extracted and separated by denaturing PAGE followed by GelRed staining. RNA was quantified with respect to the input. L = 10% input. GST-baits and pulled-down proteins are indicated with asterisks.

## RanGTP-dependent nuclear import of uS11 and Fap7

uS11 co-enriched with the 90S isolated via Noc4-TAP (*Figure 2B*). We investigated how uS11 is transported to the nuclear compartment prior to its incorporation into the 90S. In budding yeast, most r-proteins are transported into the nucleus by multiple import receptors including Kap121/Pse1, Kap123, Kap104, Sxm1 and Nmd5 (*Rout et al., 1997*; *Schlenstedt et al., 1997*; *Sydorskyy et al., 2003*).

To analyze uS11 nuclear uptake in vivo, we uncoupled its import process from incorporation into the 90S. On the mature 40S subunit the C-terminus of uS11 is deeply embedded within the rRNA framework between helix 23 and the 3' end of 18S rRNA (*Figure 3B*). We fused GFP to the C-terminus of uS11 with the aim of preventing its incorporation into the 90S. uS11-GFP localized to the nucleus indicating that the fusion protein was competent to interact with nuclear import machinery, but as expected it is unable to complement the lethality of the uS11-deletion strain (*Figure 6A* and *Figure 6—figure supplement 1A*). We exploited the uS11-GFP fusion as a tool to monitor transport of uS11 to the nucleus in different importin mutants (*Figure 6A* and *Figure 6—figure supplement 1B*). Nuclear uptake of uS11-GFP was specifically impaired in the *pse1-1* mutant and *pse1-1/kap104Δ* double mutant, but not in other tested importin mutants such as *kap104Δ*, *kap123Δ*, *msn5Δ*, *kap114Δ/sxm1Δ* and *kap120Δ/sxm1Δ/nmd5Δ*. These in vivo data suggest that uS11 employs primarily Pse1 for nuclear targeting. We were unable to directly test interactions between uS11 alone and Pse1 as well as other importins in vitro, since untagged uS11, in the absence of Fap7, was aggregation-prone. However, we found that importins Pse1 and Kap104 efficiently recruited the Fap7:uS11 complex, but not Fap7 alone (*Figure 6B* and *Figure 6—figure supplement 1C*) suggesting that uS11 can function as an adaptor to transport Fap7 to the nucleus via a 'piggyback' mechanism. Fap7 alone bound to Kap123 and Msn5 (*Figure 6—figure supplement 1C*, upper panel). Very weak or no binding was observed between Fap7 and importins Kap120, Sxm1, Mtr10 and Nmd5 (*Figure 6—figure supplement 1C*, lower panel). We evaluated whether the importin:uS11:Fap7 complexes were disassembled by RanGTP in vitro. Indeed, incubation with RanGTP efficiently dissociated Fap7:uS11 from Pse1 (*Figure 6C*).

Surprisingly, incubation of the Fap7:uS11 complex with importins Kap95, Kap114, Pdr6 and Kap123, resulted in the dissociation of the Fap7:uS11 heterodimer with only uS11 remaining bound to the importin (*Figure 6—figure supplement 1C*, upper panel), suggesting overlapping sites for Fap7 and these importins for binding to uS11. However, nuclear localization of GFP-Fap7 was not altered in different importin mutants (*pse1-1*, *kap104Δ*, *pse1-1/kap104Δ*, *kap123Δ*, *msn5Δ*, *kap114Δ/sxm1Δ* and *kap120Δ/sxm1Δ/nmd5Δ*) that we tested. In addition, GFP-Fap7 was efficiently targeted to the nucleus in an uS11-depleted strain (*Figure 6—figure supplement 1D*) suggesting that uS11 and Fap7 can be independently transported by multiple importins to the nuclear compartment.

After nuclear entry, a typical importin:cargo complex interacts with RanGTP via the N-terminal region of the importin, triggering cargo release and allowing its recycling to participate in new import cycles (*Cook et al., 2007*; *Lee et al., 2005*). Consistent with Ran-cycle requirement, both GFP-Fap7 and uS11-GFP mislocalized to the cytoplasm in *prp20-1* and *rna1-1* mutants at restrictive temperatures of 37°C (*Figure 6D*). Prp20 (Ran-GEF) and Rna1 (Ran-GAP) are the GDP/GTP-exchange and GTPase-activating factors of Ran (Gsp1 in yeast), respectively (*Görlich and Kutay, 1999*; *Vetter and Wittinghofer, 2001*).

Based on the above cell-biological and biochemical studies, we suggest that Pse1 functions as the major importin for uS11. Fap7 enters the nucleus either bound directly to Kap123, Msn5 and Kap104 or indirectly to Pse1 via uS11. Consistent with importin-facilitated entry, disruption of the Ran gradient precludes nuclear uptake of both uS11 and Fap7.

## Fap7:uS11 facilitates RanGTP-dependent release of eS26 from Pse1

Previously, we reported that an importin:eS26 complex is inefficiently dissociated upon incubation with RanGTP in vitro (*Schütz et al., 2014*). Instead, Tsr2 was able to unload eS26 from the importin:eS26 complex in a RanGTP-independent manner (*Schütz et al., 2014*). Since co-overexpression of Fap7 and uS11 compensated Tsr2-requirement in vivo, we wondered whether the Fap7:uS11 complex is able to release eS26 from a preformed Pse1:eS26 complex. To test this, we incubated a GST-Pse1:eS26 complex with the Fap7:uS11 complex, and evaluated eS26 release by Western blotting (*Figure 7A*). These analyses revealed that Fap7:uS11 did not release eS26 from Pse1. Instead, Fap7:

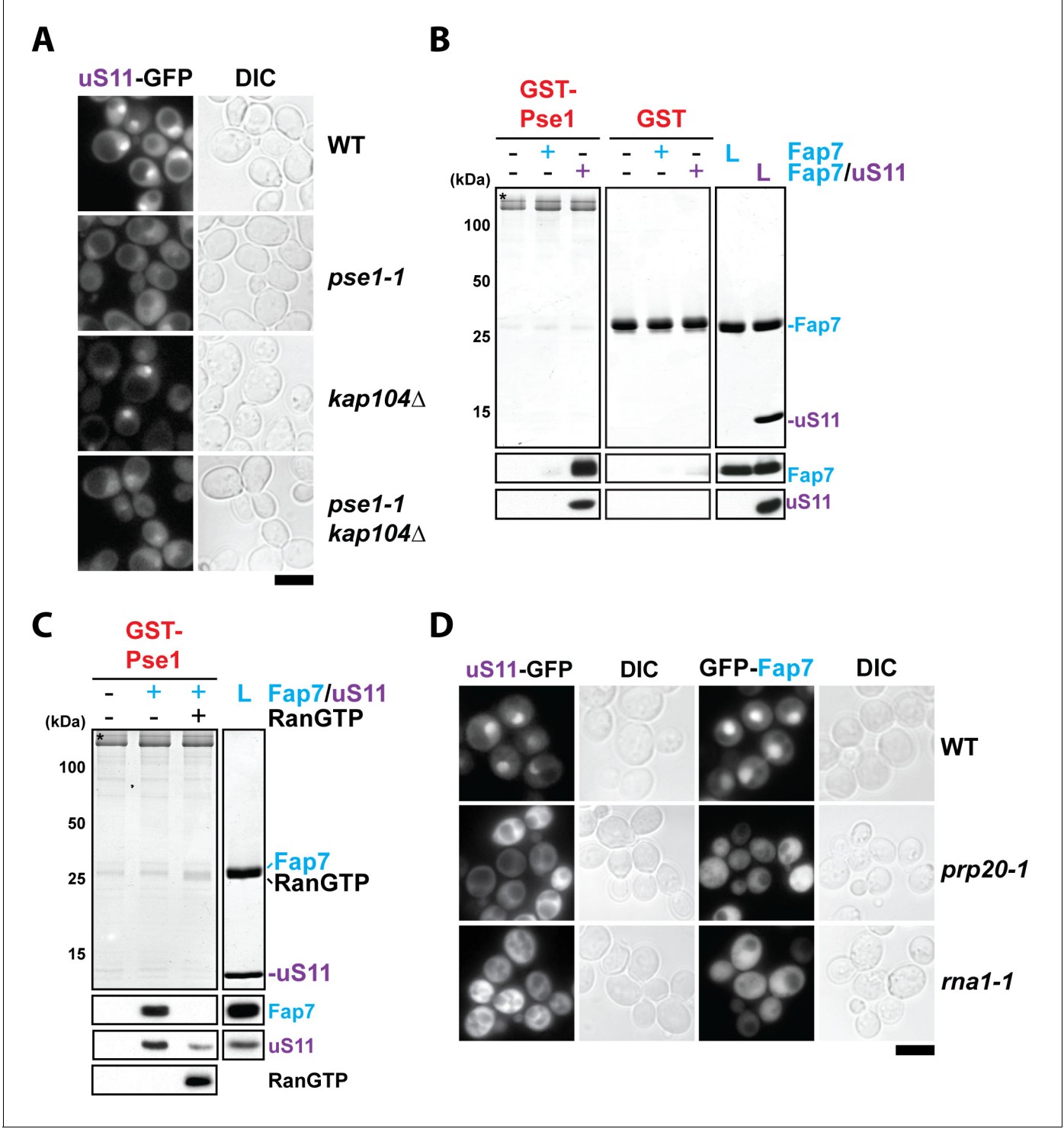

**Figure 6.** RanGTP-dependent nuclear import of uS11 and Fap7. (**A**) Nuclear uptake of uS11-GFP is impaired in *pse1-1* and *pse1-1 kap104Δ* mutants. Strains expressing uS11-GFP were grown in synthetic media at 25°C to mid-log phase. *Ts*-mutant strains were then shifted to 37°C for 4 hr and localization of uS11-GFP was analyzed by fluorescence microscopy. Scale bar = 5 μm. (**B**) Fap7:uS11, but not Fap7, interacts with Pse1. Recombinant GST-Pse1 or GST alone were immobilized on Glutathione Sepharose beads and incubated with buffer, purified Fap7 or Fap7:uS11 for 1 hr. After washing away unbound proteins, beads were eluted and analyzed by SDS-PAGE followed by Coomassie Blue staining and Western blotting. L = 10% input. GST-baits are indicated with asterisks. RanGTP (His6-Gsp1Q71L-GTP) efficiently releases Fap7:uS11 from Pse1. GST-Pse1:Fap7:uS11 complexes immobilized on Glutathione Sepharose were incubated with either buffer alone or with RanGTP for 10 min and then analyzed as described above. (**D**) Nuclear uptake of uS11-GFP and GFP-Fap7 is impaired in *prp20-1* and *rna1-1* mutants. Strains expressing uS11-GFP or GFP-Fap7 were grown in

*Figure 6 continued on next page*

*Figure 6 continued*

synthetic media at 25°C to mid-log phase. *Ts*-mutant strains were then shifted to 37°C for 4 hr and localization of uS11-GFP and GFP-Fap7 was analyzed by fluorescence microscopy. Scale bar = 5 μm.

The following figure supplement is available for figure 6:

**Figure supplement 1.** Nuclear import of uS11 and Fap7.

uS11 was recruited to the GST-Pse1:eS26 resin (*Figure 7A*, lane 3). Intriguingly, eS26 was efficiently released after incubating this resin with RanGTP (*Figure 7B*, compare lanes 3 and 5). These data suggest that the GST-Pse1:eS26 complex becomes sensitive to RanGTP-dependent disassembly possibly due to the recruitment of the Fap7:uS11 complex. To test this, we treated the GST-Pse1: eS26 with the Fap7:uS11[3R] mutant complex that is impaired in binding eS26 (*Figures 7B* and *3C*). We then incubated the resin with RanGTP and evaluated eS26 release by Western blotting. These studies revealed that, unlike the wild-type Fap7:uS11 complex, treatment with the mutant Fap7:uS11[3R] complex did not render GST-Pse1:eS26 susceptible to RanGTP disassembly (*Figure 7B*, compare lanes 5 and 7). These data support the idea that recruitment of Fap7:uS11 to the Pse1: eS26 complex through tertiary contacts renders eS26 release from Pse1 RanGTP sensitive.

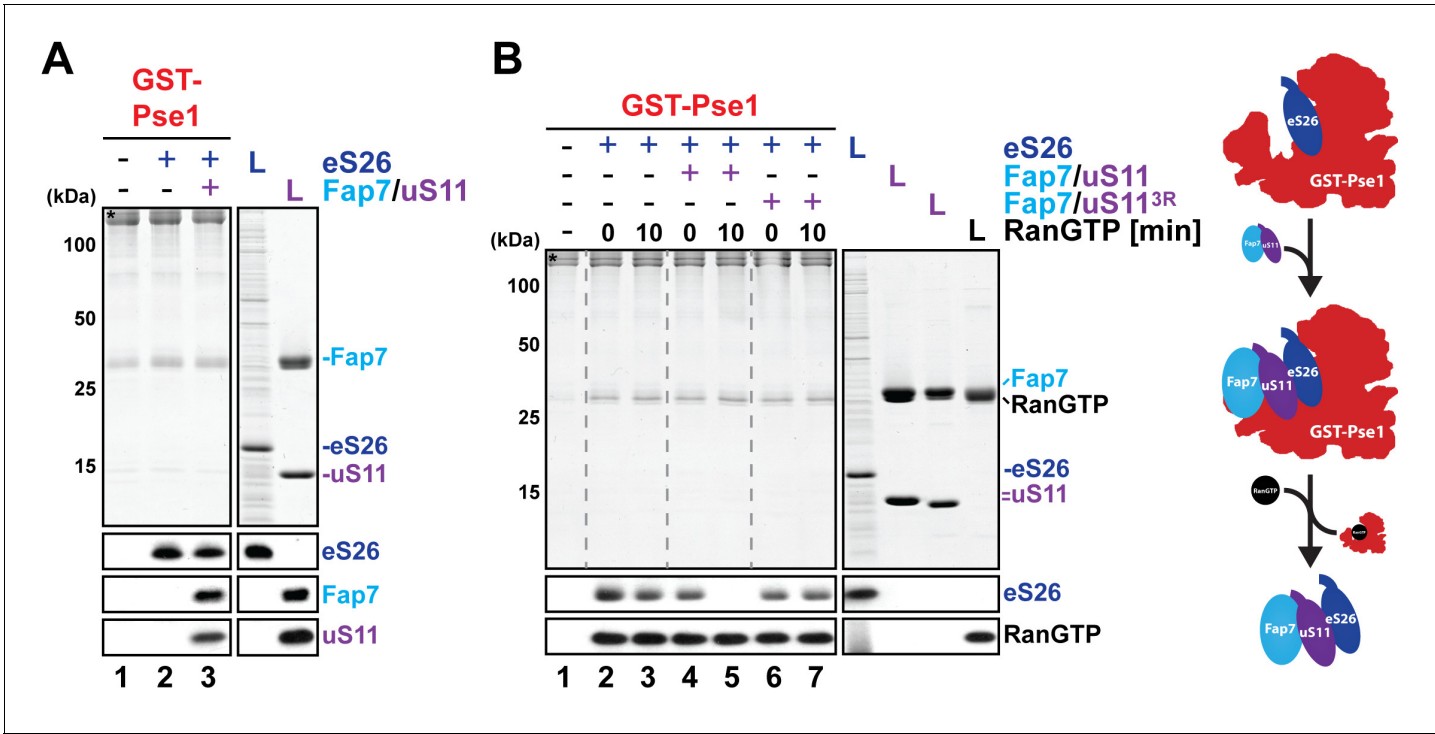

**Figure 7.** Fap7:uS11 facilitates RanGTP-dependent release of eS26 from Pse1. (**A**) eS26 and Fap7:uS11 interact with Pse1. Recombinant GST-Pse1 was immobilized on Glutathione Sepharose beads and incubated with buffer or eS26 alone. After washing away unbound proteins, beads were further incubated in presence of buffer or Fap7:uS11 for 1 hr and washed again. Beads were then eluted and analyzed by SDS-PAGE followed by Coomassie Blue staining and Western blotting. L = 10% input. GST-baits are indicated with asterisks. (**B**) Fap7:uS11 facilitates RanGTP-dependent release of eS26. Immobilized GST-Pse1 was incubated first with eS26 alone (left panel). After washing away unbound proteins, beads were further incubated with RanGTP alone or RanGTP together with either Fap7:uS11 or Fap7:uS11[3R] for indicated time points and washed again. Beads were then eluted and analyzed as described above. Right panel shows a schematic for how Fap7:uS11 facilitates RanGTP-dependent release of eS26 from Pse1.

## Discussion

Every 30 ms, a growing budding yeast cell assembles a functional ribosome by precisely targeting 79 different r-proteins to their cognate RNA binding site (*Warner, 1999*). The majority of these r-proteins need to be imported into the nucleus and then safely delivered to the ribosome assembly site in nucleolus. A challenge for the assembly machinery is to ensure that stoichiometric levels of r-proteins are incorporated into every pre-ribosome, prior to the irreversible nuclear export step. Here, a genetic screen unveiled an essential energy-dependent mechanism that delivers neighboring r-proteins eS26 and uS11 to the 90S. We show that the ATPase Fap7 promotes tertiary interactions between eS26 and uS11 and then concurrently assembles the eS26:uS11 subcomplex into the pre-ribosome. In this way, prefabrication of an r-protein subcomplex guarantees that stoichiometric levels of uS11 and eS26 are integrated into the 90S.

### Fap7 prefabricates an uS11:eS26 subcomplex for ribosome assembly

We found that r-proteins eS26 and uS11 depend upon each other for their incorporation into the 90S. Fap7-depletion compromises uS11 stability in vivo, and precludes eS26 recruitment to the 90S. Similarly, Tsr2-depletion compromises eS26 stability in vivo and precludes uS11 recruitment to the 90S. 40S pre-ribosomes assembled in Fap7- and Tsr2-depleted cells that lack both uS11 and eS26 efficiently escape nuclear proofreading, and are transported to the cytoplasm. These incompletely assembled subunits engage with mature 60S subunits to form 80S-like particles, but fail to undergo final 20S pre-rRNA processing and accumulate in the cytoplasm (*Schütz et al., 2014*; *Strunk et al., 2012*). Our findings are consistent with proteomic analyses that revealed an underrepresentation of eS26 and uS11 in these 80S-like particles that accumulate upon Fap7-depletion (*Strunk et al., 2012*).

The Fap7:uS11 complex with the assistance of RanGTP bypasses Tsr2-requirement in extracting eS26 from its importin in vitro. Further, co-overexpression of FAP7 and uS11 bypasses the functional requirement of Tsr2 to safely transfer the r-protein to the 90S in vivo. These two findings provided critical clues to unveil the mechanistic interdependency of eS26 and uS11 incorporation into the 90S. uS11, only when bound to Fap7, recruits eS26 suggesting that Fap7 renders uS11 competent to load eS26. This interaction requires conserved tertiary contacts between uS11 and eS26 found on the mature 40S subunit and is critical for bypassing Tsr2-requirement. The mechanism of Fap7-mediated pre-organization of an uS11:eS26 r-protein subcomplex for ribosome assembly is essential and distinct from the non-essential symportin Syo1. Syo1 does not promote tertiary contacts between r-proteins uL5 and uL11. Instead, it directly binds to both the r-proteins at two distinct sites and then mediates their co-import into the nucleus for 5S rRNP assembly (*Calviño et al., 2015*; *Kressler et al., 2012*).

eS26, when bound to Tsr2, was not recruited to the Fap7:uS11 complex, suggesting that eS26 needs to be released from Tsr2 before engaging with the Fap7:uS11 complex. How eS26 dissociates from Tsr2 remains unclear. While Fap7 ATPase activity is not required to form a Fap7:uS11:eS26 complex, ATP hydrolysis is critical to load uS11:eS26 onto H23 rRNA. We therefore suggest that Fap7 scaffolds the uS11:eS26 complex formation and its ATPase activity regulates integration of this r-protein subcomplex into the 90S. Our findings are consistent with in vitro studies that showed that Fap7 could efficiently load uS11 onto H23 rRNA in an ATP-dependent manner (*Loc'h et al., 2014*). However, in that study, the cellular compartment in which uS11 loading onto the pre-ribosome occurs was not investigated. Our data support a model in which Fap7 loads the pre-formed uS11: eS26 complex during early nucleolar assembly of the 90S pre-ribosome. The essential nature and enzymatic activity requirement for its function distinguishes Fap7 from another dedicated chaperones such as Acl4. The Acl4 chaperone system captures the r-protein uL4 in the cytoplasm for Kap104-mediated import. Subsequently, the interaction of the eukaryotic-specific uL4 extension with the pre-ribosomal surface is thought to trigger energy-independent disassembly of the Acl4:RpL4 complex and allow for incorporation of uL4 (*Pillet et al., 2015*; *Stelter et al., 2015*).

### Two distinct mechanisms extract eS26 from the import machinery

Given that the *RPS26* knockout is lethal in yeast, whereas Tsr2-depleted cells are viable but severely growth impaired, implicates another mechanism that unloads eS26 from the import machinery and safely delivers the r-protein to the 90S. Our data demonstrate that the Fap7:uS11 complex and

RanGTP are components of this alternative mechanism that sustains viability of the $P_{GAL1}$-*TSR2* mutant. In vitro, Fap7:uS11 interacts with the importin:eS26 complex and renders the importin:eS26 complex sensitive to RanGTP-dependent disassembly. This pathway involves tertiary contacts between uS11 and eS26, since mutations that disturb interactions between Fap7:uS11 and eS26 are unable to release eS26 from its importin and to compensate Tsr2-requirement in vivo. How Fap7:uS11 recruitment renders the importin:eS26 complex sensitive to RanGTP is unclear. It is tempting to speculate that an element within eS26 may be responsible to prevent its premature RanGTP-dependent release from its importin. Recruitment of Fap7:uS11 might relieve this negative regulation and permit RanGTP-dependent release. In general, such a mechanism could couple cargo release from importins with its transfer to an interaction partner and/or final functional destination in the nucleus.

Based on the observation that co-overexpression of Fap7 and uS11 efficiently suppresses Tsr2-depletion phenotypes, we suggest that, in wild-type cells, there are two pools of the Fap7:uS11 complex. The major pool seems to perform the essential duty of organizing a Fap7:uS11:eS26 subcomplex and incorporating it into the 90S. Only a small pool of the Fap7:uS11 complex is available for its adjunct role in RanGTP-dependent unloading of eS26 directly from the importin and is therefore limiting in Tsr2-depleted cells. Overexpression of Fap7 and uS11 would increase free Fap7:uS11 complex levels to perform the adjunct duty of extracting eS26 from its importin and compensating the absence of Tsr2. We propose two mechanisms that operate to extract eS26 from the import machinery. The major RanGTP insensitive mechanism involves Tsr2 that extracts eS26 from the importin:eS26 complex, and forms a stable Tsr2:eS26 complex. Subsequently, eS26 is released from Tsr2, and then recruited to the preformed Fap7:uS11 complex. The alternative mechanism employs the preformed Fap7:uS11 complex for eS26 extraction from its importin. In this case, the Fap7:uS11 interacts with eS26 when bound to its importin, and renders eS26 release RanGTP-sensitive. Importantly, both mechanisms converge to form a Fap7:uS11:eS26 complex for incorporation into the 90S.

## Prefabricating r-protein subcomplexes for correct ribosome assembly?

A major task for the assembly machinery is to precisely target thousands of r-proteins every second to developing pre-ribosomes. It is apparent that mechanisms must exist that rapidly and, at the same time, correctly incorporate different r-proteins into their cognate rRNA binding sites, and thereby prevent production of a non-functional ribosome. A striking structural feature of eukaryotic r-proteins is that they cluster on the surface of the ribosome (*Figure 3—figure supplement 1*) through a network of ~140 tertiary contacts (*Poirot and Timsit, 2016*). The function(s) of these intricate conserved inter-connections, how and when they are established has remained unclear. Prefabricating r-protein subcomplexes through these tertiary contacts prior to their assembly provides one mechanism to reduce the molecularity of the incorporation process and ensure stoichiometric levels of r-proteins reach their rRNA binding sites. Thus, Fap7 may represent a family of factors that pre-organize native-like r-protein subcomplexes and minimize production errors during eukaryotic ribosome formation.

## Materials and methods

### Yeast strains and plasmids

The *S. cerevisiae* strains used in this study are listed in *Supplementary file 1A*. Genomic disruptions, C-terminal tagging and promoter switches at genomic loci were performed as described previously (*Janke et al., 2004*; *Longtine et al., 1998*; *Puig et al., 2001*). Preparation of media, yeast transformations and genetic manipulations were performed according to established procedures.

Plasmids used in this study are listed in *Supplementary file 1B*. Details of plasmid construction will be provided upon request. All recombinant DNA techniques were performed according to established procedures using *E. coli* XL1 blue cells for cloning and plasmid propagation. Point mutations in *FAP7* and *RPS14A (uS11)* were generated using the QuikChange site-directed mutagenesis kit (Agilent Technologies, Santa Clara, CA). All cloned DNA fragments and mutagenized plasmids were verified by sequencing.

## Fluorescence in situ hybridization and microscopy

Localization of 20S pre-rRNA was analyzed using a Cy3-labeled oligonucleotide probe (5'-Cy3-ATG CTC TTG CCA AAA CAA AAA AAT CCA TTT TCA AAA TTA TTA AAT TTC TT-3') that is complementary to the 5' portion of ITS1 as previously described (*Faza et al., 2012*).

Pre-40S subunit export, monitored by localization of uS5-GFP and localization of GFP-tagged Fap7 and uS11 was performed as previously described (*Altvater et al., 2014*; *Faza et al., 2012*; *Pertschy et al., 2007*; *Milkereit et al., 2001*). Cells were visualized using a DM6000B microscope (Leica, Germany) equipped with a HCX PL Fluotar 63×/1.25 NA oil immersion objective (Leica, Germany). Images were acquired with a fitted digital camera (ORCA-ER; Hamamatsu Photonics, Japan) and Openlab software (Perkin-Elmer, USA). All Cell-biological studies were performed at least on three different occasions and in triplicates; >99% of cells showed the reported phenotypes in a sample size of >1000 cells.

## RNA and protein extraction from yeast

WCEs were prepared by alkaline lysis of yeast cells as previously described (*Kemmler et al., 2009*). To analyze RNAs after whole cell lysis, RNA was extracted with Phenol-Chlorofom-Isoamylalcohol from yeast lysate and precipitated in isopropanol. RNA pellets were washed with 80% ethanol and finally resuspended in 20 μl water. One microgram of total RNA was separated on a 1.2% Agarose/formaldehyde gel for 1.5 hr at 200 V. For Northern analysis, rRNAs were blotted onto a Hybond-XL (Amersham, UK) membrane by capillary transfer and probed for 18S (5'-CATGCATGGCTTAATC TTTGAGAC), 20S (5'-GGTTTTAATTGTCCTATAACAAAAGC) and 25S rRNA (5'-TGCCGCTTCAC TCGCCGTTAC) using radioactively labeled probes. rRNAs were detected using phosphoimaging screens (GE Healthcare).

Tandem affinity purifications (TAP) of pre-ribosomal particles were carried out as previously described (*Faza et al., 2012*). Calmodulin-eluates were separated on NuPAGE 4–12% Bis-Tris gradient gels (Invitrogen) and visualized by either Silver staining or Western analyses using indicated antibodies. All biochemical purifications were performed at least on three different occasions and in triplicates.

Western analyses were performed as previously described (*Kemmler et al., 2009*). The following primary antibodies were used in this study: α-Fap7/uS11 (1:2,000; this study), α-Tsr2/S26 (1:3,000; *Schütz et al., 2014*), α-uS7 (yeast Rps5)(1:4,000; Proteintech Group Inc., Chicago, IL), α-uS3 (yeast Rps3)(1:3,000; M. Seedorf, University of Heidelberg, Heidelberg, Germany); α-TAP (CBP) (1:4,000; Thermo Scientific, Rockford, IL), α-Pno1 (1:10,000; K. Karbstein, Scripps Research Institute, Jupiter, FL), α-Dim1 (1:10,000; K. Karbstein, Scripps Research Institute, Jupiter, FL), α-Nob1 (1:500; Proteintech Group Inc., Chicago, IL), α-Tsr1 (1:10,000; K. Karbstein, Scripps Research Institute, Jupiter, FL). The secondary HRP-conjugated α-rabbit and α-mouse antibodies (Sigma-Aldrich, USA) were used at 1:1,000–1:5,000 dilutions. Protein signals were visualized using Immun-Star HRP chemiluminescence kit (Bio-Rad Laboratories, Hercules, CA) and captured by Fuji Super RX X-ray films (Fujifilm, Japan).

## Recombinant protein expression

All recombinant proteins were expressed in *E. coli* BL21 cells by IPTG induction (20°C for 16 hr). His6-tagged and GST fusion proteins were affinity-purified by batch purification in standard buffer (20 mM Hepes pH 7.5, 500 mM NaCl, 2 mM EDTA, 10% glycerol) using Ni-NTA Agarose (GE Healthcare) or Glutathione Sepharose (GE Healthcare), respectively. GST-tagged importins and RanGTP (His6-Gsp1Q71L-GTP) were expressed and purified as previously described (*Fries et al., 2007*; *Maurer et al., 2001*; *Solsbacher et al., 1998*).

## ATPase and ATP binding assays

The ATPase activity assays were essentially performed as described (*Loc'h et al., 2014*). This coupled-enzyme assay relies on the formation of ADP upon ATP hydrolysis by Fap7. Generated ADP molecules are coupled to β-NADH oxidation via the action of pyruvate kinase (PK) and lactate dehydrogenase (LDH). The ATPase activity of Fap7 was then monitored over time by the decrease in β-NADH absorbance at 340 nm. Reactions were performed in 96-well plates and 20 μl total volume containing 5–10 μM recombinant protein, 20 mM Hepes pH7.5, 100 mM NaCl, 10% glycerol, 1 mM DTT, 2 mM MgCl2, 3 mM ATP, 8 mM Phosphoenolpyruvate, 1.2 mM NADH, each 20 U/ml PK/LDH.

ATP-binding assays were performed as described (*Manikas et al., 2016*; *Montpetit et al., 2012*) using fluorescently labeled mant-ATP (2'-/3'-O-(N'-methylanthraniloyl) adenosine-5'-O- triphosphate). Reactions were performed in 96-well plates and 100 µl total volume containing 10 µM recombinant protein and 1 µM mant-ATP in 20 mM Hepes pH7.5, 250 mM NaCl, 5 mM MgCl2. The reactions were incubated for 10 min at 30°C and then excited at 355 nm with a xenon lamp. Emission spectra were recorded between 385 and 600 nm at a resolution of 5 nm using a Clariostar plate reader (BMG Labtech, Germany). All assays were performed on at least three different occasions and in triplicates.

### In vitro interaction studies

Recombinant and GST-tagged Fap7:uS11 complexes were either co-expressed in *E. coli* or reconstituted in vitro and then immobilized in binding buffer (20 mM Hepes pH 7.5, 250 mM NaCl, 5 mM MgCl2, 0.2% Tween-20) on Glutathione Sepharose first for 30 min at 4°C followed by 15 min at 30°C as described (*Loc'h et al., 2014*). When using ATP or AMP-PNP, Fap7 was briefly pre-incubated with 1 mM nucleotide prior to incubation with uS11. To test binding to eS26, the beads were incubated with *E. coli* lysate containing recombinant eS26FLAG. Beads were then washed thrice and eluted in two-fold LDS-sample buffer by incubating at 70°C. Pulled-down proteins were then separated by SDS-PAGE and visualized by Coomassie Blue staining and/or by Western analysis using antibodies against eS26 and Fap7:uS11. For binding assays including 18S helix 23 rRNA (5'-GGG AGU AUU CAA UUG UCA GAG GUG AAA UUC UUG GAU UUA UUG AAG ACU-3', Microsynth, Switzerland), an additional incubation step was performed for 1 hr at 4°C followed by a quick wash and Phenol-Chloroform-extraction. Bound RNAs were analyzed on a 10% acrylamide-8M urea gel and stained with GelRed (Biotium, Fremont, CA).

The in vitro binding studies between recombinant eS26FLAG, Fap7:uS11 and yeast importins as GST-fusion proteins were performed as previously described (*Aitchison et al., 1996*; *Seedorf and Silver, 1997*; *Solsbacher et al., 1998*). All binding assays were performed in presence of 100 µl competing *E.coli* lysate.

To dissociate the GST-Pse1:eS26FLAG complex, pre-immobilized GST-Pse1:eS26FLAG complexes were incubated with buffer alone or 1.5 µM His6-Fap7:uS11 and/or 1.5 µM RanGTP (His6-Gsp1Q71L-GTP) at 4°C (*Schütz et al., 2014*). Samples were withdrawn at 0 and 10 min and analyzed by Coomassie Blue staining and Western analysis. All binding studies were performed on at least three different occasions and in triplicates.

## Acknowledgements

We are grateful to M Peter, G Schlenstedt, J Woolford, K Karbstein, and M Seedorf for generously sharing strains and antibodies, and N Schäuble, M Hondele and K Weis for help with ATPase assays. We thank all members of the Panse laboratory, in particular J Thorner UC Berkeley, for enthusiastic discussions, J Petkowski for structure-guided analysis, M Altvater for microscopy, and the Institute of Medical Microbiology UZH for continued support. V G Panse is supported by grants from the Swiss National Science Foundation, NCCR in RNA and Disease, ETH Zurich, Novartis Foundation, Olga Mayenfisch Stiftung and a Starting Grant Award (EURIBIO260676) from the European Research Council.

## Additional information

### Funding

| Funder | Grant reference number | Author |
| --- | --- | --- |
| Eidgenössische Technische Hochschule Zürich | | Ute Fischer Yiming Chang Vikram G Panse |
| University of Zurich | | Vikram G Panse |
| Schweizerischer Nationalfonds zur Förderung der Wissenschaftlichen Forschung | | Vikram G Panse |

| Novartis Foundation | | Vikram G Panse |
| NCCR RNA & Disease | | Vikram G Panse |
| Olga Mayenfisch Stiftung | | Vikram G Panse |
| European Research Council | EURIBIO260676 | Vikram G Panse |

The funders had no role in study design, data collection and interpretation, or the decision to submit the work for publication.

### Author contributions
CP, Conception and design, Acquisition of data, Analysis and interpretation of data, Drafting or revising the article; SS, Acquisition of data, Analysis and interpretation of data, Drafting or revising the article; UF, YC, Conception and design, Acquisition of data, Analysis and interpretation of data; VGP, Conception and design, Analysis and interpretation of data, Drafting or revising the article

### Author ORCIDs
Vikram G Panse, http://orcid.org/0000-0001-7950-5746

# Additional files

### Supplementary files
• Supplementary file 1. Yeast strains and plasmids used in this study. (A) Yeast strains used in this study. (B) Plasmids used in this study.

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
