## [Decision Letter]

Thank you for submitting your article "Prefabrication of a ribosomal protein subcomplex essential for eukaryotic ribosome formation" for consideration by *eLife*. Your article has been favorably evaluated by James Manley (Senior Editor) and three reviewers, one of whom, Alan G Hinnebusch (Reviewer #1), is a member of our Board of Reviewing Editors. The following individual involved in review of your submission has agreed to reveal their identity: John Woolford (Reviewer #2).

The reviewers have discussed the reviews with one another and the Reviewing Editor has drafted this decision to help you prepare a revised submission.

Summary:

This paper examines the role of the conserved ATPase Fap7 in proper assembly of 40S ribosomal proteins uS11 and eS26 into the 90S pre-ribosome of yeast. They began by isolating Fap7 and uS11 as overexpression suppressors of the lethality and assembly defect of eS26 conferred by depletion of Tsr1, the escortin that unloads eS26 from its import receptor in the nucleus by a Ran-GTP-independent mechanism. They showed that efficient suppression occurs only when Fap7 and uS11 are co-overexpressed, and obtained evidence that Fap7 and uS11 function in concert with eS26 and Tsr1 in proper assembly of uS11 and eS26 into ribosomes. They reconstituted in vitro a GST-Fap7:uS11:eS26 complex and showed that it requires amino acid contacts between uS11 and eS26 found in the mature ribosome, as well as the ATPase activity of Fap7, and then demonstrated that this complex could be loaded onto rRNA helix 23, again dependent on ATPase activity, thus providing strong evidence that Fap7 orchestrates formation of an uS11:eS26 complex, and that its ATPase activity facilitates loading of this prefabricated r-protein subcomplex onto h23 rRNA during ribosome biogenesis. (How eS26 dissociates from Tsr2 to enable this process remains unclear, as this does not appear to be an activity of the uS11:eS26 complex.) Other results demonstrate that the Fap7:uS11 complex and RanGTP are components of an alternative mechanism that can operate in cells lacking Tsr1 to unload eS26 from the nuclear import machinery. They showed that Fap7:uS11 binding renders the importin:eS26 complex sensitive to RanGTP, which it ordinarily is not, again dependent on native uS11:eS26 contacts. They propose that in wild-type cells, the Fap7:uS11 complex performs the essential duty of organizing a Fap7:uS11:eS26 subcomplex and incorporating it into the 90S, but when overexpressed can also perform an adjunct role in RanGTP-dependent unloading of eS26 from the importin thereby compensating for Tsr2-depletion.

Essential revisions:

1) It is important to show by Western analysis that the failure of overexpressed catalytically dead fab1-7 to confer suppression of the Tsr2 depletion is not a trivial result of its failure to accumulate in cells.

2) It is important to attempt to demonstrate, e.g. by coIP analysis, that eS26 accumulates in a complex with its importin in Tsr2-depleted cells, which is reversed by co-overexpression of Fap7 and uS11.

3) The relative ratios of H23 to eS26 in lanes 8-13 of Figure 5 must be quantified.

4) It is important to investigate whether a tagged from of eS26 is capable of interacting with H23 in the absence of S11, a possibility strongly suggested by the data in Figure 5.

*Reviewer #1:*

This paper examines the role of the conserved ATPase Fap7 in proper assembly of 40S ribosomal proteins uS11 and eS26 into the 90S pre-ribosome of yeast. They began by isolating Fap7 and uS11 as overexpression suppressors of the lethality and assembly defect of eS26 conferred by depletion of Tsr1, the escortin that unloads eS26 from its import receptor in the nucleus by a Ran-GTP-independent mechanism. They showed that efficient suppression occurs only when Fap7 and uS11 are co-overexpressed, and obtained evidence that Fap7 and uS11 function in concert with eS26 and Tsr1 in proper assembly of uS11 and eS26 into ribosomes. They reconstituted in vitro a GST-Fap7:uS11:eS26 complex and showed that it requires amino acid contacts between uS11 and eS26 found in the mature ribosome, as well as the ATPase activity of Fap7, and then demonstrated that this complex could be loaded onto rRNA helix 23, again dependent on ATPase activity, thus providing strong evidence that Fap7 orchestrates formation of an uS11:eS26 complex, and that its ATPase activity facilitates loading of this prefabricated r-protein subcomplex onto h23 rRNA during ribosome biogenesis. (How eS26 dissociates from Tsr2 to enable this process remains unclear, as this does not appear to be an activity of the uS11:eS26 complex.) Other results demonstrate that the Fap7:uS11 complex and RanGTP are components of an alternative mechanism that can operate in cells lacking Tsr1 to unload eS26 from the nuclear import machinery. They showed that Fap7:uS11 binding renders the importin:eS26 complex sensitive to RanGTP, which it ordinarily is not, again dependent on native uS11:eS26 contacts. They propose that in wild-type cells, the Fap7:uS11 complex performs the essential duty of organizing a Fap7:uS11:eS26 subcomplex and incorporating it into the 90S, but when overexpressed can also perform an adjunct role in RanGTP-dependent unloading of eS26 from the importin thereby compensating for Tsr2-depletion.

General critique: The results are a strong combination of genetics and biochemistry, and of in vivo and in vitro analyses, which mainly support the main conclusions of the paper, and the paper is largely well written. While pertaining to only a single aspect of ribosome biogenesis, the findings will likely have broader implications for other ribosome assembly steps. There are two experimental shortcomings that should be redressed and several unclear passages that should be revised.

Subsection “Co-overexpression of Fap7 and the r-protein uS11 bypasses in vivo requirement for Tsr2”, second paragraph: It wasn't stated whether the high-copy plasmids were subcloned and re-tested to confirm that the FAP7 and uS11 genes were responsible for the suppression observed. This would be required.

Subsection “Fap7 ATPase activity is required to bypass Tsr2 requirement”, last paragraph: It is necessary to show that fap7-2 protein accumulates to levels comparable to that of WT Fap7 to justify the claim that loss of catalytic activity underlies the loss of suppression.

*Reviewer #2:*

Ribosomal proteins are abundant, basic proteins, many of which contain extensions predicted to have dis-ordered structures. To efficiently and accurately integrate these r proteins into assembling ribosomes in eukaryotes, cells must overcome several challenges: nascent r proteins must be folded properly to avoid aggregation, chaperoned to prevent inappropriate interactions with other proteins or RNAs, imported into the nucleus, and properly configured into nascent ribosomes in equal numbers. Most r proteins are essential; if one of them fails to assemble in a timely fashion or assemble properly, assembly of that ribosomal subunit fails.

This manuscript describes an elegant combination of molecular genetic experiments in vivo and biochemical assays in vitro leading the authors to suggest a much more detailed "prefabrication" mechanism used by yeast to overcome these challenges, to deliver r proteins S11 and S26 into adjacent positions in small ribosomal subunits.

Previous work revealed that assembly factor Tsr2 helps S26 assemble and assembly factor Fap7 helps S11 assemble. Here it is shown that Fap7 and S11 are imported into the nucleus separately or as a complex, released from import factors using RAN, then bind to S26 and deliver it to pre-ribosomes using ATP hydrolysis by Fap7. Most of the time, S26 is released from import factors by Tsr2. When this occurs, it is not clear how Tsr2 dissociates from S26 before S26 can bind to Fap7-S11, but this step appears to be RAN independent. When Tsr2 is not used to dissociate S26 from import factors, the Fap7-S11 complex can release S26 from import factors in a RAN-dependent fashion. The interesting point is that assembly of S11 and S26 is interdependent, brought together with Fap7, presumably in the nucleus, before they are inserted into pre-ribosomes using ATP hydrolysis by Fap7.

In my opinion the experiments support the authors' model. The manuscript could be improved by the following:

1) Introduction, second paragraph: I do not think that 200 assembly factors are required for large subunit assembly; rather, closer to 80. Can the authors double-check this number?

2) Introduction, last paragraph, and Discussion, first paragraph: the statements imply that all r proteins assemble before nuclear export. The authors state that particles escape nuclear proofreading in the absence of Fap7. Perhaps a detail for this work, but they should clarify that a few other r proteins assemble in the cytoplasm in wild type cells, but that S11 and S26 are among the majority that assemble in the nucleus.

3) The authors might add to the Discussion:

How does their work extend and/or contradict previous work describing how Fap7 helps S11 assemble with rRNA, e.g., the work of Loc'h et al.

In addition maybe briefly compare and contrast their ideas re S11 and S26 with published models for incorporation of r protein L4 by its chaperone, e.g., where or how Fap7 recognizes and binds S11.

4) Subsection “Two distinct mechanisms extract eS26 from the import machinery”, last paragraph: to clarify the discussion, can the authors state upfront that the existence of major and minor pools of the Fap7-S11 complex and their respective roles in assembly are a model based on results in the paper, in particular based on the suppression of Tsr2 depletion phenotypes by extra copies of both and the S11 and FAP7 genes?

*Reviewer #3:*

This manuscript describes the coordinated assembly of the ribosomal proteins uS11 and eS26 by Fap7 into the pre-ribosome. The work builds on previous work from the Panse lab that investigated the role of Tsr2 in unloading eS26 from importins after import into the nucleus. In that work, the authors described a RanGTP-independent pathway requiring Tsr2 but also argued that there was a second pathway of unloading. Here, the authors show that a complex of Fap7-uS11 can act in a RanGTP-dependent fashion to release eS26. Experiments are well executed, the data are very clean and well displayed and the logic is largely well laid out. However, I have several concerns regarding the work that should be addressed.

1) The authors write "Co-overexpression of FAP7 and uS11 bypassed Tsr2-requirement in extracting eS26 from its importin and safely transferring the r-protein to the 90S." This is not actually shown but is based on extrapolation from in vitro results. Does eS26 accumulate in a complex with its importin in Tsr2 depleted cells and is this relived by overexpression of Fap7 and uS11? This should be tested in vivo by co-IP. This point is important not just to justify this statement but also because this is a major conclusion of the work that is based almost entirely on in vitro analysis.

2) Figure 5, the authors claim that the loading of uS11 onto H23 is dependent on Fap7 and recruitment of eS26. The dependence on Fap7 is quite nice but the retention of H23 seems to correlate closely with eS26 levels. Is H23 binding to uS11 as claimed or to eS26? The relative ratios of H23 to eS26 in lanes 8-13 should be quantified. The authors must also include controls of eS26 alone with H23.

3) If uS11 needs Fap7 for its stability, as suggested by Figure 2, the model for independent import of uS11 seems unlikely. When uS11 accumulates in the cytoplasm in pse1-1 cells, is it associated with Fap7? Do they co-IP? If not, how is it imagined that uS11 is imported without Fap7 to stabilize it? Similarly, when Fap7 accumulates in the cytoplasm in RanGTP cycle mutants, is it associated with uS11?

4) Figure 6. It's not clear what the significance is of uS11 interacting with many different importins. These proteins are typically highly negatively charged. Is it possible these simply represent nonspecific electrostatic interactions driven by the high in vitro protein concentrations?

---

## [Author Response]

[…]

*Essential revisions:*

*1) It is important to show by Western analysis that the failure of overexpressed catalytically dead fab1-7 to confer suppression of the Tsr2 depletion is not a trivial result of its failure to accumulate in cells.*

This is an important control to exclude the possibility that a catalytically inactive Fap7 is unstable in vivo, and therefore unable to rescue the observed phenotypes associated with Tsr2-depletion. To this end, Western analyses of whole cell lysates derived from Tsr2-depleted cells overexpressing Fap7 and the catalytically inactive fap7-2 was performed. These studies show that overexpressed catalytically inactive Fap7-2 protein accumulates to levels comparable to that upon WT Fap7 overexpression (Figure 4, left panel).

*2) It is important to attempt to demonstrate, e.g. by coIP analysis, that eS26 accumulates in a complex with its importin in Tsr2-depleted cells, which is reversed by co-overexpression of Fap7 and uS11.*

As requested, we have attempted to address by coIP analyses the specific concern raised by reviewer 3, point 1.

First, we analyzed eS26-FLAG protein levels in WT, Tsr2-depleted cells and Tsr2-depleted cells co-overexpressing *FAP7* and *uS11*. As expected, in whole cell extracts (WCE) eS26-FLAG levels were strongly reduced upon Tsr2-deleption, which were restored upon *FAP7* and *uS11* co-overexpression (Figure 8).

Author response image 1.(**A**) Whole cell extracts (WCE) were prepared from indicated strains and subjected to Western analysis using antibodies directed against Pse1 and Kap123. uS7 served as loading control. (**B**) eS26-FLAG was isolated from indicated strains. Equal levels of eS26-FLAG were loaded and separated on a SDS-gel and subjected to Western analysis using the indicated antibodies.**DOI:**
http://dx.doi.org/10.7554/eLife.21755.013

Next, we immuno-precipitated S26-FLAG from these cells (Figure 8). We loaded equal amounts of eS26-FLAG isolated from each these cells on a SDS-gel and performed Western analyses to monitor importin (Pse1 and Kap123) co-enrichment (Figure 8). However, we were unable to co-enrich detectable levels of Pse1 and Kap123 in these IPs. One possible explanation could be that at steady state only a small amount of eS26, below the detection levels of our Western analyses, is bound to its importin during transfer to the nucleus. Alternatively, it is possible that the importin:eS26 complexes fall apart upon cell lysis.

In response to the specific sentence in the Discussion that was criticized by reviewer 3 (point 1), we would like to clarify our idea regarding Tsr2 function. Our model presented in Schütz et al., 2014 *eLife* was based on a combination of in vitro and in vivo findings. In that study, we showed that importin:eS26 complexes, unlike typical importin:cargo complexes, were inefficiently dissociated by RanGTP in vitro. Instead, we found that Tsr2, without the assistance from RanGTP, efficiently releases eS26 from its importin, and prevents its aggregation in vitro. in vivo, Tsr2-depletion renders eS26 susceptible to proteolysis, and precludes its incorporation into the 90S pre-ribosome. Based on these and other data, we proposed a model in which Tsr2 triggers the release of eS26 from its importin in the nucleus and therefore directly couples the process of eS26 nuclear import and transfer to the 90S.

In the current study, we revealed how Fap7 and uS11 can compensate Tsr2-requirement, both in vitro and in vivo. Specifically:

A) Fap7:uS11 recruitment to the Pse1:eS26 complex rendered the resulting Pse1:eS26:uS11:Fap7 complex sensitive to RanGTP-dependent disassembly in vitro.

B) Fap:uS11 overexpression rescued eS26 levels in vivo, and ensured that eS26 is safely transferred to the 90S in the Tsr2-depletion mutant.

We agree with reviewer 3 that our sentence in the Discussion section "Co-overexpression of FAP7 and uS11 bypassed Tsr2-requirement in extracting eS26 from its importin and safely transferring the r-protein to the 90S." is not precisely phrased. Therefore, we have corrected this to: “The Fap:uS11 complex with the assistance of RanGTP bypasses Tsr2-requirement in extracting eS26 from its importin in vitro. Co-overexpression of *FAP7* and *uS11* bypasses the functional requirement of Tsr2 to safely transfer the r-protein to the 90S in vivo.”.

*3) The relative ratios of H23 to eS26 in lanes 8-13 of Figure 5 must be quantified.*

*4) It is important to investigate whether a tagged from of eS26 is capable of interacting with H23 in the absence of S11, a possibility strongly suggested by the data in Figure 5.*

We have repeated the binding assays presented in Figure 5 (see Figure 5). In particular, we have included controls to directly compare and quantify H23 rRNA levels bound to GST-uS11 and GST-uS11:eS26 complexes. These studies revealed that, in the presence of Fap7 and ATP, nearly identical levels of H23 rRNA are recruited to GST-uS11 and GST-uS11:eS26 complexes (Figure 5, compare lanes 5 and 7 and quantification). Further, as requested, we tested whether a tagged version of eS26 (GST-eS26) interacts with H23 rRNA in absence of uS11. We could not detect any interaction between GST-eS26 and H23 RNA (Figure 5, lane 1-2).

Together, these data strongly support the idea that H23 rRNA specifically interacts with the GST-uS11:eS26 complex through uS11. These experiments have been included in the Results subsection “Fap7 ATPase activity organizes and recruits the uS11:eS26 subcomplex to helix 23 of 18S rRNA”.

*Reviewer #1:*

*[…]General critique: The results are a strong combination of genetics and biochemistry, and of in vivo and in vitro analyses, which mainly support the main conclusions of the paper, and the paper is largely well written. While pertaining to only a single aspect of ribosome biogenesis, the findings will likely have broader implications for other ribosome assembly steps. There are two experimental shortcomings that should be redressed and several unclear passages that should be revised.*

*Subsection “Co-overexpression of Fap7 and the r-protein uS11 bypasses* in vivo *requirement for Tsr2”, second paragraph: It wasn't stated whether the high-copy plasmids were subcloned and re-tested to confirm that the FAP7 and uS11 genes were responsible for the suppression observed. This would be required.*

This was indeed performed. We have now included a statement that *FAP7* and *uS11* genes were subcloned into high-copy plasmids, and their ability to rescue phenotypes associated with Tsr2-depletion was re-tested and confirmed (Results subsection “Co-overexpression of Fap7 and the r-protein uS11 bypasses in vivo requirement for Tsr2”, second paragraph).

*Subsection “Fap7 ATPase activity is required to bypass Tsr2 requirement”, last paragraph: It is necessary to show that fap7-2 protein accumulates to levels comparable to that of WT Fap7 to justify the claim that loss of catalytic activity underlies the loss of suppression.*

We thank the reviewer for pointing out this important control. Please see our response above to the Essential revisions section (point 1).

*Reviewer #2:*

*[…]In my opinion the experiments support the authors' model. The manuscript could be improved by the following:*

*1) Introduction, second paragraph: I do not think that 200 assembly factors are required for large subunit assembly; rather, closer to 80. Can the authors double-check this number?*

We apologize for this mistake, and have corrected the number. Indeed, the 60S assembly pathway requires ~80 assembly factors (Introduction, second paragraph).

*2) Introduction, last paragraph, and Discussion, first paragraph: the statements imply that all r proteins assemble before nuclear export. The authors state that particles escape nuclear proofreading in the absence of Fap7. Perhaps a detail for this work, but they should clarify that a few other r proteins assemble in the cytoplasm in wild type cells, but that S11 and S26 are among the majority that assemble in the nucleus.*

We agree that our statements appear to imply that all r-proteins are assembled in the nucleus. This is clearly not the case. We have now rephrased these sentences (Introduction, fourth paragraph and Discussion, first paragraph), and clearly state that we refer to only those r-proteins that need to enter the nucleus for incorporation into the developing pre-ribosome.

*3) The authors might add to the Discussion:*

*How does their work extend and/or contradict previous work describing how Fap7 helps S11 assemble with rRNA, e.g., the work of Loc'h et al.*

*In addition maybe briefly compare and contrast their ideas re S11 and S26 with published models for incorporation of r protein L4 by its chaperone, e.g., where or how Fap7 recognizes and binds S11.*

We thank the reviewer for this comment that has helped to improve our Discussion. We have now included a comparison for our study with those to the studies by Loc’h et al. and added a section in which our findings are put into context of how uL4 is incorporated (subsection “Fap7 prefabricates an uS11:eS26 subcomplex for ribosome assembly”, last paragraph).

*4) Subsection “Two distinct mechanisms extract eS26 from the import machinery”, last paragraph: to clarify the discussion, can the authors state upfront that the existence of major and minor pools of the Fap7-S11 complex and their respective roles in assembly are a model based on results in the paper, in particular based on the suppression of Tsr2 depletion phenotypes by extra copies of both and the S11 and FAP7 genes?*

We have now clearly stated that our model is based on our genetic suppression data (Discussion subsection “Two distinct mechanisms extract eS26 from the import machinery”, last paragraph).

*Reviewer #3:*

*[…]1) The authors write "Co-overexpression of FAP7 and uS11 bypassed Tsr2-requirement in extracting eS26 from its importin and safely transferring the r-protein to the 90S." This is not actually shown but is based on extrapolation from* in vitro *results.*

Our model regarding the function of Tsr2 presented in Schütz et al., 2014 *eLife* is based on a combination of in vitro and in vivo findings. In that study, we showed that importin:eS26 complexes, unlike typical importin:cargo complexes, were inefficiently dissociated by RanGTP in vitro. Instead, we found that Tsr2 without the assistance from RanGTP, efficiently releases eS26 from its importin, and prevents its aggregation in vitro. in vivo Tsr2-depletion renders eS26 susceptible to proteolysis, and precludes its incorporation into the 90S pre-ribosome. Based on these and other data, we proposed that Tsr2 triggers the release of eS26 from its importin in the nucleus and therefore directly couples the process of eS26 nuclear import and transfer to the 90S.

Here, we reveal how Fap7 and uS11 can compensate Tsr2-requirement both in vitro and in vivo. Specifically:

A) Fap7:uS11 recruitment to the Pse1:eS26 complex rendered the resultant Pse1:eS26:uS11:Fap7 complex sensitive to RanGTP-dependent disassembly in vitro.

B) Fap:uS11 overexpression rescued eS26 levels in vivo, and ensured that eS26 is safely transferred to the 90S in the Tsr2-depletion mutant.

We agree with the reviewer that our sentence in the Discussion section "Co-overexpression of FAP7 and uS11 bypassed Tsr2-requirement in extracting eS26 from its importin and safely transferring the r-protein to the 90S." does not precisely reflect these findings. Therefore, we have corrected the sentence to: “The Fap:uS11 complex with the assistance of RanGTP bypasses Tsr2-requirement in extracting eS26 from its importin in vitro. Co-overexpression of FAP7 and uS11 bypasses the functional requirement of Tsr2 to safely transfer the r-protein to the 90S in vivo.”

*Does eS26 accumulate in a complex with its importin in Tsr2 depleted cells and is this relived by overexpression of Fap7 and uS11? This should be tested* in vivo *by co-IP.*

Please see our response above to the Essential revisions section (point 2).

*This point is important not just to justify this statement but also because this is a major conclusion of the work that is based almost entirely on* in vitro *analysis.*

We politely disagree with this reviewer’s opinion regarding the major conclusion of this work. The main finding of this study is that the ATPase Fap7 brings together uS11 and eS26 interactions through native-like tertiary contacts and its ATPase activity is required to integrate a pre-formed uS11:eS26 complex into the 90S. This ensures stoichiometric integration of uS11 and eS26 into the 90S pre-ribosome. Different lines of genetic, cell-biological and in vitro */*in vivo biochemical evidences support this conclusion. Specifically:

A) Tsr2-depletion compromises eS26 stability and precludes uS11 integration into the 90S. Fap7-depletion compromises uS11 stability and precludes eS26 integration into the 90S. These in vivo data demonstrate an interdependence of eS26 and uS11 for their incorporation into the 90S.

B) Together with the above data and the specific genetic interaction between Tsr2 and Fap7:uS11 (suppression data) led us to show that Fap7 organizes a uS11:eS26 complex through tertiary contacts and that its ATPase activity is required to load this complex onto H23 rRNA. The importance of native-like contacts between uS11:eS26, and the importance of Fap7 ATPase activity to load H23 rRNA were validated by genetic, cell-biological and in vivo biochemical studies.

*2) Figure 5, the authors claim that the loading of uS11 onto H23 is dependent on Fap7 and recruitment of eS26. The dependence on Fap7 is quite nice but the retention of H23 seems to correlate closely with eS26 levels. Is H23 binding to uS11 as claimed or to eS26? The relative ratios of H23 to eS26 in lanes 8-13 should be quantified. The authors must also include controls of eS26 alone with H23.*

We thank the reviewer for pointing out this important control. Please see our response to points 3 and 4 in the Essential revisions above.

*3) If uS11 needs Fap7 for its stability, as suggested by Figure 2, the model for independent import of uS11 seems unlikely. When uS11 accumulates in the cytoplasm in pse1-1 cells, is it associated with Fap7? Do they co-IP? If not, how is it imagined that uS11 is imported without Fap7 to stabilize it?*

As requested, we have immuno-precipitated uS11-GFP from WT and *pse1-1* cells (Figure 9). These biochemical studies show that uS11-GFP can co-enrich Fap7. It is unlikely that all uS11-GFP in the cytoplasm is bound to Fap7, since nuclear targeting of GFP-Fap7 is not impaired in a *pse1-1* mutant.

Author response image 2.(**A**) Wild-type and *pse1-1* strains expressing uS11-GFP were grown in synthetic media at 25°C to mid-log phase and then shifted to 37°C for 4 h. uS11-GFP was then immuno-precipitated and subjected to Western analysis.(**B**) Wild-type and *prp20-1* strains expressing Fap7-TAP were grown in synthetic media at 25°C to mid-log phase and then shifted to 37°C for 4 h. Fap7-TAP was then purified and subjected to Western analysis.**DOI:**
http://dx.doi.org/10.7554/eLife.21755.014

*Similarly, when Fap7 accumulates in the cytoplasm in RanGTP cycle mutants, is it associated with uS11?*

As requested, we have immuno-precipitated Fap7-TAP from WT and *prp20-1* cells (Figure 9). These biochemical studies show that Fap7 can co- enrich uS11.

Newly synthesized uS11, after emerging from the translating ribosome, is likely to be stabilized by cytoplasmic factors before being captured by importins. Yeast cells deficient for the ribosome associated NAC and SSB-RAC chaperone systems that associate with newly synthesized polypeptides accumulate several ribosomal protein aggregates (Koplin et al., 2010). NAC and SSB-RAC chaperone systems might bind uS11 before their transport to the nucleus. Upon arrival in the nuclear compartment interaction with Fap7 stabilizes uS11 and prepares the r-protein together with eS26 for assembly.

*4) Figure 6. It's not clear what the significance is of uS11 interacting with many different importins. These proteins are typically highly negatively charged. Is it possible these simply represent nonspecific electrostatic interactions driven by the high* in vitro *protein concentrations?*

We could not test interactions between uS11 and the different importins, since uS11 is aggregation-prone when not bound to Fap7. All interaction studies between importins and Fap7 and Fap7:uS11 complexes are well-established binding assays in the nuclear transport field. These assays are always carried out in the presence of competing *E. coli* lysates to minimize non-specific interactions.

Budding yeast employs 11 importins to target diverse cargos to the nucleus. In this study, we have tested interactions between Fap7:uS11 complex and Fap7 with all 11 importins. Amongst the 11 importins, we observe differences in binding. For e.g. Sxm1, Kap120, Mtr10 and Nmd5 hardly interact with Fap7 as well as the Fap7:uS11 complex. In contrast, Msn5 efficiently binds to Fap7, but rather weakly with the Fap7:uS11 complex. Finally, only Pse1 and Kap104 binds to the Fap7:uS11 complex. Therefore, we do not think that these interactions are non-specific electrostatic interactions.